# Recent Advances in Electrochemical Sensors for Formaldehyde

Yufei Yang [1], Yuanqiang Hao [1,2] (ID), Lijie Huang [1], Yuanjian Luo [2], Shu Chen [2,*] (ID), Maotian Xu [1] and Wansong Chen [3,*] (ID)

1   College of Chemistry and Chemical Engineering, Shangqiu Normal University, Shangqiu 476000, China; yangyufei1986@163.com (Y.Y.); haoyuanqiang@aliyun.com (Y.H.); 13213221725@163.com (L.H.); xumaotian@sqnu.edu.cn (M.X.)
2   Key Laboratory of Theoretical Organic Chemistry and Functional Molecule of Ministry of Education, School of Chemistry and Chemical Engineering, Hunan University of Science and Technology, Xiangtan 411201, China; 15197155233@163.com
3   College of Chemistry and Chemical Engineering, Central South University, Changsha 410017, China
*   Correspondence: chenshu@hnust.edu.cn (S.C.); chenws@csu.edu.cn (W.C.)

**Abstract:** Formaldehyde, a ubiquitous indoor air pollutant, plays a significant role in various biological processes, posing both environmental and health challenges. This comprehensive review delves into the latest advancements in electrochemical methods for detecting formaldehyde, a compound of growing concern due to its widespread use and potential health hazards. This review underscores the inherent advantages of electrochemical techniques, such as high sensitivity, selectivity, and capability for real-time analysis, making them highly effective for formaldehyde monitoring. We explore the fundamental principles, mechanisms, and diverse methodologies employed in electrochemical formaldehyde detection, highlighting the role of innovative sensing materials and electrodes. Special attention is given to recent developments in nanotechnology and sensor design, which significantly enhance the sensitivity and selectivity of these detection systems. Moreover, this review identifies current challenges and discusses future research directions. Our aim is to encourage ongoing research and innovation in this field, ultimately leading to the development of advanced, practical solutions for formaldehyde detection in various environmental and biological contexts.

**Keywords:** formaldehyde; electrochemical sensor; enzyme; electrocatalyst; electrochemical probe

## 1. Introduction

Formaldehyde, represented by the chemical formula $CH_2O$, stands as a fundamental organic compound and the simplest aliphatic aldehyde. The carbon–oxygen double bond imparts significant polarity to formaldehyde, making it highly reactive in chemical reactions. Additionally, formaldehyde exhibits the typical characteristics of an aldehyde group, facilitating a range of specific reactions. Formaldehyde finds extensive applications in various industrial sectors, including resin production, antimicrobial agents and disinfectants, textile processing, pharmaceutical manufacturing, biomedical research, furniture, and building materials [1–5]. However, the extensive use of formaldehyde has raised alarms due to its role as a significant indoor air pollutant, classified as the third-largest by the World Health Organization in 2004 [6].

Additionally, formaldehyde is a crucial bioactive molecule intricately involved in various biological processes. Within plant organisms, formaldehyde can be generated through pathways involving methyl transfer and demethylation [7]. As part of chemical communication, plants release formaldehyde as a response to environmental stress [8]. Furthermore, formaldehyde serves as a secondary metabolite produced during the synthesis of specific compounds, potentially endowed with defensive or protective functions. Confronted with external pressures such as pathogen infections, climate variations, or damage, plants may produce formaldehyde as part of their stress response mechanism. In summary,

the generation of formaldehyde within plant organisms involves multifaceted processes encompassing physiological activities, metabolic pathways, and interactions with the external environment, potentially playing a crucial role in plant growth, development, and adaptability [5]. In animals and the human body, endogenous formaldehyde holds significance with various physiological functions, primarily generated through metabolic reactions. These reactions are especially orchestrated by enzymes such as semicarbazide-sensitive amine oxidase (SSAO) and cytochrome P450 in coordinated demethylation reactions. SSAO, a copper-dependent amine oxidase, catalyzes the deamination of compounds like methylamine, resulting in the production of formaldehyde [9]. This enzyme is widespread in organs such as the brain, heart, and liver. Cytochrome P450, on the other hand, participates in enzymatic reactions aimed at breaking down foreign substances, thereby aiding in the production of formaldehyde to eliminate extraneous materials. Importantly, cellular organelles like the endoplasmic reticulum play a pivotal role in the formation of formaldehyde, including processes like succinate-enhanced formaldehyde accumulation. Additionally, various components in the cell nucleus or cytoplasm can transform into formaldehyde, contributing to biological methylation and other essential processes [10]. Moreover, formaldehyde plays a crucial role in cellular signal transduction by covalently binding with proteins and nucleic acids, thereby regulating their activity and stability [11]. This modification process is implicated in various biological processes, including cell growth, differentiation, and apoptosis, underscoring the key role of formaldehyde in maintaining cellular signal balance and functional stability [12].

However, exposure to an environment with excessive formaldehyde and elevated levels of formaldehyde in the human body can have severe consequences. Such exposure can lead to discomfort in the eyes, respiratory problems, skin irritation, and adverse effects on the nervous and immune systems [13]. The concentration of formaldehyde in human blood varies between 10 μM and 100 μM, and higher accumulation of formaldehyde is associated with organ aging [14]. Elevated formaldehyde concentrations can intensify apoptotic activity or reduce mitotic activity in cells [15]. Research has also identified that excessive formaldehyde exposure can trigger various diseases, including chronic inflammation, fetal development issues, cardiovascular diseases, leukemia, and nasopharyngeal cancer [16–19]. Therefore, the detection of formaldehyde is crucial and has sparked extensive research. Accurately monitoring formaldehyde levels is essential for protecting human health and preventing potential risks in the environment. Moreover, research in this field is continually expanding, exploring more advanced and sensitive detection technologies to gain a more comprehensive understanding of the distribution and impact of formaldehyde in different environments. Through an exploration of the physicochemical properties of formaldehyde, a variety of detection methods have been developed, including gas chromatography [20], high-performance liquid chromatography (HPLC) [21,22], infrared spectroscopy [23], Raman spectroscopy [24,25], colorimetric methods [26,27], fluorescence spectroscopy [28–32], mass spectrometry [33,34], electrochemical methods [35–41], and quartz crystal microbalance [42,43], among others. Each of these methods possesses its own unique characteristics. For instance, chromatography methods offer high accuracy but involve complex operational procedures and relatively slow detection speeds, limiting real-time monitoring capabilities. Spectroscopic methods, on the other hand, provide rapid response times, particularly fluorescence analysis, which allows for in situ real-time monitoring. However, spectroscopic methods often require the use of derivatization or fluorescent probe molecules, necessitating the specific design and synthesis of responsive molecules. Comprehensive reviews have been conducted to provide an overview of the relevant research on these methods [14,44–49].

Here, we focus on electrochemical methods for formaldehyde sensing. Electrochemical techniques are renowned for their high sensitivity, selectivity, and real-time monitoring capabilities, making them ideal tools for formaldehyde detection. Electrochemical formaldehyde sensors can be categorized into two types. One type is semiconductor-based formaldehyde alert sensors. These sensors, known for their simplicity, low cost, and com-

pact size, have been a mainstream choice in the market and have received extensive research attention. Their basic principle involves the reaction of formaldehyde molecules with adsorbed oxygen on the surface of semiconductor materials, leading to the flow of electrons or holes, thereby reducing the thickness of the hole accumulation layer and electron depletion layer, subsequently altering conductivity or changing voltammetric characteristics and surface potential to achieve a response to formaldehyde. This category of sensors has been widely studied, and several review papers have summarized recent research progress in this field [50–56]. Another category comprises electrochemical sensors based on solution reactions. These sensors primarily involve the chemical oxidation of formaldehyde at the electrode surface, resulting in a current or potential response. These sensors operate under milder reaction conditions and, by combining different electrode materials and sensing modes, can develop various high-performance formaldehyde sensors. However, to the best of our knowledge, numerous related studies having been reported (Scheme 1), there is currently no comprehensive review paper that summarizes and analyzes these sensors. Therefore, this review will provide an overview of this category of electrochemical sensors, introducing the types of electrode materials used, elucidating the fundamental principles of their detection, summarizing different sensor configurations, and subsequently providing examples of their applications. Finally, we will discuss the current limitations of these sensors and propose potential avenues for future research, aiming to drive the continuous development of electrochemical methods for formaldehyde detection.

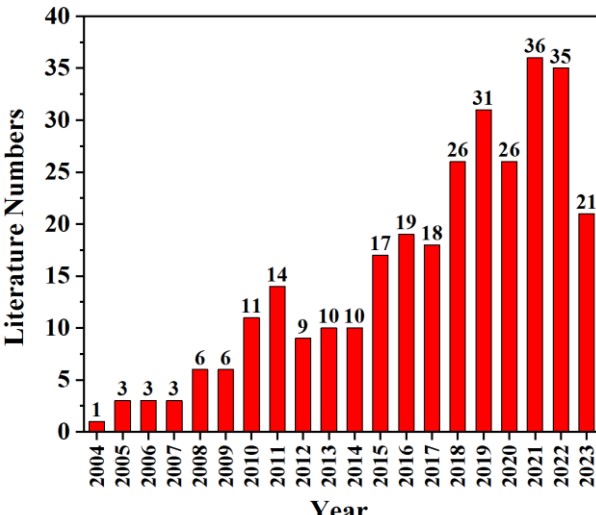

**Scheme 1.** The number of papers published on formaldehyde electrochemical sensors in the past 20 years. Data were retrieved from Web of Science using the search terms "Electrochemical Sensors" and "Formaldehyde" as search topics. Search date: 5 January 2024.

## 2. Sensing Mechanisms of Electrochemical Formaldehyde Sensors

Electrochemical formaldehyde sensors based on solution reactions can be primarily categorized into three types based on their electrode reaction mechanisms. The first category encompasses enzymatic formaldehyde sensors, which rely on biological enzymes for formaldehyde detection. The second category comprises electrochemical sensors that utilize electrocatalysts (i.e., inorganic metals or metal oxides) to catalyze the oxidation of formaldehyde. Finally, the third category encompasses electrochemical sensors derived from specific chemical molecules. These distinct sensor types offer various advantages and applications, contributing to the versatility and effectiveness of formaldehyde detection methods.

The operational principle of enzymatic formaldehyde electrochemical sensors relies on the highly specific catalytic activity of enzymes. These sensors primarily consist of a working electrode modified with a specific enzyme, typically formaldehyde dehydrogenase (FDH), which is specialized in catalyzing the oxidation of formaldehyde. The electrode's operational process is illustrated in Figure 1. In the presence of the co-reactant $NAD^+$

(nicotinamide adenine dinucleotide), FDH catalyzes the oxidation of formaldehyde to form formic acid (HCOOH). Simultaneously, this reaction involves the conversion of $NAD^+$ to NADH as $NAD^+$ accepts electrons from the formaldehyde oxidation process. Subsequently, electrons are transferred from NADH to the electrode, resulting in the oxidation of NADH back to $NAD^+$. This electron transfer generates an electrical current, which can be correlated with the formaldehyde concentration and measured to quantify its presence. The key characteristic of this enzymatic formaldehyde electrochemical sensor is its exceptional specificity. Enzymes demonstrate a high level of selectivity for formaldehyde, resulting in minimal interference from other substances. This property enables the sensor to accurately detect formaldehyde while remaining unaffected by other compounds.

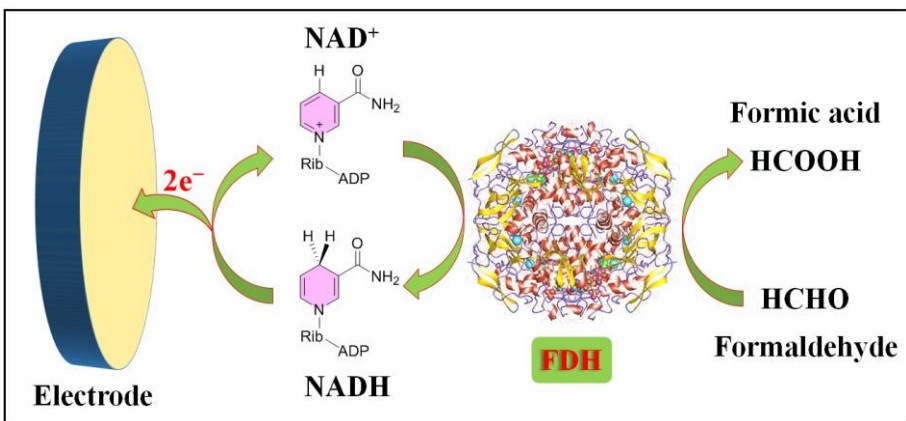

**Figure 1.** Formaldehyde electrochemical sensors rely on bioenzymes.

Another crucial approach to developing formaldehyde sensors involves the electrocatalytic oxidation of formaldehyde using different electrocatalysts (Figure 2). With the advancement of nanomaterial synthesis and characterization techniques, an increasing number of such electrochemical formaldehyde sensors have emerged over the past decade. These sensors primarily employ various electrocatalysts, including elemental metals, metal alloys, metal oxides, hydroxides, heterogeneous materials, and non-metallic electrocatalysts. Depending on the type and properties of the electrocatalysts, which encompass variations in the redox potentials of metal species, the electronic conductivity of materials, and their adsorption capacity for formaldehyde and oxidation intermediates, the process of catalyzing formaldehyde oxidation, as well as the resulting catalytic products, can differ. Oxidation products may include formic acid, carbon monoxide, carbon dioxide, and others. However, these sensors rely on the direct oxidation of the target analyte, formaldehyde. Consequently, other reducible substances may introduce interference. Therefore, precise control of the electrocatalyst's structure and reaction activity is of paramount importance.

Formaldehyde exhibits versatile chemical properties due to its active carbonyl group and hydrogen atoms. Firstly, the carbonyl group's high electrophilicity allows it to react with organic compounds such as thiols and amines, resulting in various chemical reactions, including aldehyde–amine condensation, 2-aza-Cope rearrangement, Pictet–Spengler reaction, and Hantzsch reaction. Secondly, formaldehyde possesses reducing properties and can reduce specific metal ions to their anionic forms while oxidizing itself to formic acid. Additionally, formaldehyde can react with bases to form methanol salts. These distinct chemical reactivity characteristics of formaldehyde have paved the way for the development of electrochemical sensors. There have been reports of electrochemical sensors based on formaldehyde derivatization reactions [57–59]. Typically, these sensors utilize specific derivatization reagents that selectively react with formaldehyde, leading to the formation of probe molecules with new electrochemical properties upon interaction with formaldehyde (Figure 3). This approach enables highly selective formaldehyde detection and has been extensively applied in fluorescence-based assays. However, its application in electrochemical detection is relatively limited, indicating the potential for further exploration in this area.

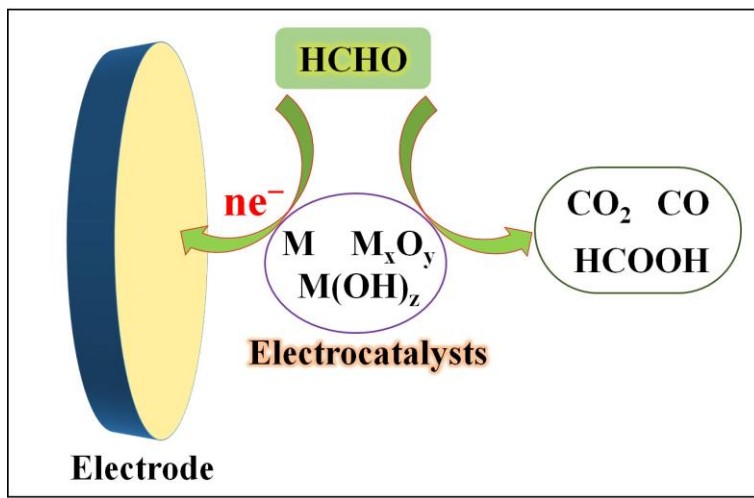

**Figure 2.** Formaldehyde electrochemical sensors rely on electrocatalysts.

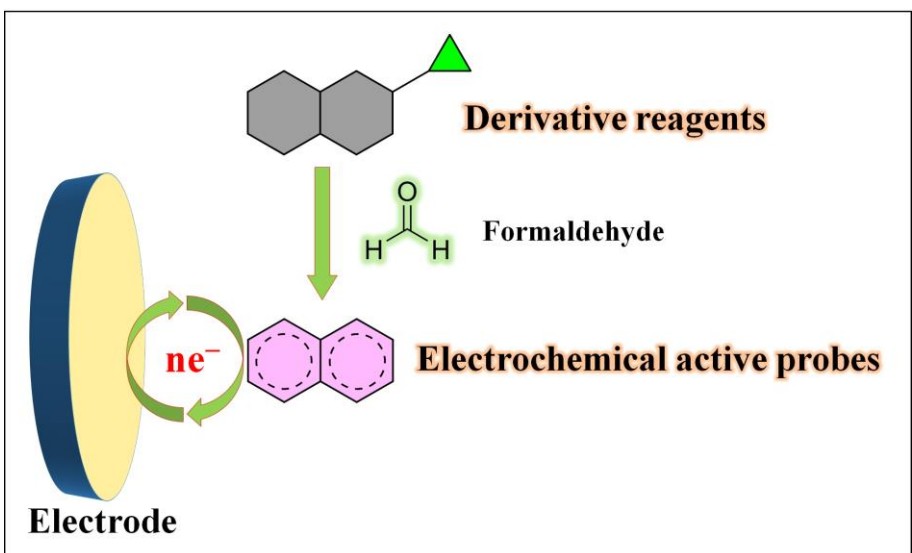

**Figure 3.** Formaldehyde electrochemical sensors rely on derivative reagents.

### 3. Electrochemical Sensors for Formaldehyde

*3.1. Electrochemical Sensors Rely on Bioenzymes*

Enzyme-based formaldehyde electrochemical sensors are one of the earliest reported categories of formaldehyde sensors. The core components of these sensors are enzymes. Currently, two main types of enzymes are used, namely formaldehyde dehydrogenase and alcohol oxidase (AOX). Below, we will provide individual introductions for each of them (Table 1).

**Table 1.** Electrochemical sensors rely on bioenzymes.

| Electrode Materials | Signal Mode | Dynamic Range | Detection Limit | Applications | Ref. |
|---|---|---|---|---|---|
| Pt/FDH | Amperometry | 0–6 vppm | 0.3 vppm | Gas sensing | [60] |
| SPEs/POs-EA/FDH | Amperometry | 0.030–1.5 $mg \cdot mL^{-1}$ | 30 $ng \cdot mL^{-1}$ | -- | [61] |
| Au/CdS/FDH@Nylon | Amperometry | 0.05–1 $\mu g \cdot mL^{-1}$ | 41 $ng \cdot mL^{-1}$ | -- | [62] |
| Graphite/PVI-Os/DPH/FDH | Amperometry | 0.05–0.5 mM | 32 $\mu M$ | -- | [63] |

**Table 1.** *Cont.*

| Electrode Materials | Signal Mode | Dynamic Range | Detection Limit | Applications | Ref. |
|---|---|---|---|---|---|
| PTFE/graphite/FDH | Amperometry | 0–15 ppm | 0.03 ppm | -- | [64] |
| GCE/FDH@FSM8.0/Nylon | Amperometry | 1.2–617 μM | 1.1 μM | -- | [65] |
| Au electrodes/Dextran@FDH | Conductometry | 10–200 mM | -- | -- | [66] |
| SPEs/CNTs | Amperometry | 0.1–100 μM | -- | -- | [67] |
| Au/Nafion@FDH | Amperometry | 0.1–10 ppm | 0.016 ppm | Fish, Mackerel | [68] |
| GCE/poly(GMA-co-MTM)@FDH/PPyfilm | Amperometry | 3.3–100 μM | 0.15 μM | Water | [69] |
| Interdigitated gold electrodes/Nafion@FDH-GA | Conductometry | 0–10 mM | 18 μM | Water | [70] |
| AuSPE/MWNT@PBA-FDH | Amperometry | 10 ppb to 10 ppm | 6 ppb | Urine | [71] |
| ITO/$\alpha$-Fe$_2$O$_3$/FDH | Amperometry | 0.0–0.3 mg·L$^{-1}$ | 0.02 mg·L$^{-1}$ | Juice | [72] |
| ITO/CNT-Fe$_3$O$_4$/FDH | Amperometry | 0.05–0.50 mg·L$^{-1}$ | 0.05mg·L$^{-1}$ | Juice | [73] |
| Bioanode: BP/PMG/FDH Cathode: CFP/Au NPs/PB | Amperometry Colorimetry | 0.01–0.35 mM 0.01-0.045 mM | 0.006 mM | Water | [39] |
| SPE/FDH/BSA ITO/$\alpha$-Fe$_2$O$_3$/FDH | Amperometry Colorimetry | 0.01–0.5 mg/L 0.01–0.5 mg/L | 0.03 mg/L 0.03 mg/L | Corn | [74] |
| Bioanode: FDH/PMG/BP Cathode: ITO/PB | Amperometry Colorimetry | 80 and 3000 ppb | | Gas | [38] |
| Au electrode/Au NPs/FDH | Amperometry | 0.25–2.0 mM | 0.05 mM | Water | [75] |
| SPE/ErGO/AuPd/Cys/FDH | Amperometry | 1–100 μM | 0.3 μM | Fish | [76] |
| Cu electrode/ALDHs | Amperometry | $10^{-15}-10^{-5}$ M | $10^{-15}$ M | -- | [35] |
| Gold interdigitated electrode/DEAE-dextran@lactitol/BSA@AOX | Conductometry | 0.05$-$500 mM | 0.05 mM | -- | [77] |
| SPE(Ag/AgCl)/pH transducer/poly(nBA-NAS)-AOX | Potentiometry | 0.3–316.2 mM | 0.3 mM | Shrimp | [78] |
| SPCPtEs/BSA@AOX | Amperometry | 60-460 μM | 60 μM | Histogen | [79] |
| Carbon (or gold) electrode/AOX&HRP(aq.) | Amperometry | 0–5 mM | -- | Gas | [80] |
| SPE(Ag/AgCl)/pHEMA/poly(nBA-NAS)-AOX | potentiometry | 0.5–220.0 mM | 0.1 mM | Fish | [81] |

### 3.1.1. Formaldehyde-Dehydrogenase-Based Formaldehyde Sensors

Formaldehyde dehydrogenase (FDH, EC 1.2.1.46) is an enzyme that plays a pivotal role in the metabolism of formaldehyde within living organisms. Its primary function involves catalyzing the oxidation of formaldehyde to formic acid, concurrently reducing the coenzyme nicotinamide adenine dinucleotide (NAD$^+$) to NADH. This enzymatic reaction constitutes a vital component of the cellular pathway responsible for detoxifying formaldehyde.

Leveraging its enzymatic reactivity, FDH has been employed in the development of electrochemical biosensors tailored for formaldehyde detection. As far back as 1996, Hall and colleagues engineered sensors based on FDH for the direct measurement of formaldehyde vapor in gas-phase environments [60]. Within the electrochemical reaction cell, FDH is immobilized on the surface of either a platinum (Pt) or graphite working electrode. A gas-permeable Teflon membrane effectively segregates the gas phase (containing the target gas, formaldehyde) from the internal electrolyte of the electrochemical cell. Co-reactant NAD$^+$ and electrochemical mediator 1,2-naphthoquinone-4-sulfonic acid (NQS) are introduced into the detection system. The detection mechanism hinges on the

diffusion of formaldehyde vapor into the electrochemical cell. Here, FDH catalyzes the oxidation of formaldehyde while concurrently reducing $NAD^+$ to NADH (Equation (1)). The electrochemical mediator NQS facilitates electron transfer between NADH and the electrode (Equations (2) and (3)). Notably, this sensor demonstrates a nearly linear response across formaldehyde concentrations ranging from 0 to 6 vppm, achieving a detection limit (LOD) of 0.3 vppm. Moreover, the sensor exhibits remarkable stability.

$$CH_2O + NAD^+ + 3H_2O \xrightarrow{FDH} HCO_3^- + NADH + 2H_3O^+ \tag{1}$$

$$H^+ + NADH + NQS \rightarrow NAD^+ + NQSH_2 \tag{2}$$

$$NQSH_2 \xrightarrow{electrode} NAD^+ + 2e^- + 2H^+ \tag{3}$$

Researchers have developed a plethora of formaldehyde electrochemical sensors based on FDH by combining various electrode substrate materials, electrochemical electron transfer mediators, and different methods of enzyme immobilization on the electrode surface, as summarized in Table 1. Rishpon et al. integrated a biosensor measurement device with a flow injection system and employed formaldehyde dehydrogenase (FDH) in conjunction with $Os(bpy)_2$-poly(vinylpyridine) (POs-EA) chemically modified screen-printed electrodes for sensitive detection of formaldehyde in solution [61]. Vastarella et al. introduced a novel photoelectrochemical biosensor by immobilizing FDH onto CdS nanocrystals, attaching them to a gold electrode through self-assembled monolayers [62]. Covalent enzyme immobilization enhanced sensor stability, and light-induced electron/hole recombination in the semiconductor nanoparticles triggered catalytic oxidation of formaldehyde by FDH. This biosensor exhibited high sensitivity (detection limit of 41 ppb), selectivity, and operational stability under flow conditions. Notably, it eliminated the need for $NAD^+$/NADH as a charge transfer mediator, simplifying its design and operation, making it promising for sensitive and selective formaldehyde detection in various applications. Subsequently, conductive polymer materials were employed for electrode preparation to better support FDH, maintaining its stability and activity, such as Nafion, Dextran, PPy, etc. Furthermore, electrode modifications with other inorganic materials, including graphite, carbon nanotubes, graphene, porous silicon, gold nanoparticles, etc., were found to enhance sensor performance.

Metal and bimetallic nanoparticles have been widely employed to modify electrodes, significantly enhancing their surface area and catalytic activity. Such modifications are extensively used in electrochemical sensors to improve their performance. Ratautas et al. introduced the development of a direct electron transfer (DET) biosensor for formaldehyde determination in river water [75]. The biosensor involves immobilizing formaldehyde dehydrogenase (FDH) on a gold-nanoparticle-modified gold electrode. Notably, this study achieved DET for FDH for the first time, with formaldehyde oxidation occurring at a low electrode potential of $-190$ mV vs. Ag/AgCl. The biosensor serves as a mediatorless tool for formaldehyde detection, featuring a detection limit of 0.05 mM and a linear detection range from 0.25 to 2.0 mM. It demonstrates remarkable stability and selectivity and has been successfully applied to determine added formaldehyde concentrations in river water samples. Bhatt et al. designed a biosensor utilizing a thiol-functionalized graphene nanocomposite decorated with formaldehyde dehydrogenase [76]. The sensor incorporates fern-like gold–palladium dendritic deposition on a printed electrode, enabling the detection of NADH and, consequently, formaldehyde. Direct electron transfer is achieved by lowering the oxidation potential of NADH from +0.63 V to 0.32 V vs. Ag/AgCl, eliminating the need for electron mediators. The sensor exhibits a detection limit of 0.3 μM for formaldehyde and a linear detection range between 1 μM and 100 μM when studied using chronoamperometry with an applied potential of +0.35 V vs. Ag/AgCl. The sensor demonstrates excellent recovery rates in simulated formaldehyde-spiked fish and milk samples, establishing a simple "on-site" disposable formaldehyde detection sensor. This developed

biosensor holds promise for widespread applications in the quantitative measurement of NADH and analytes related to coenzyme-associated reactions.

Combining the traditional enzyme-based bioanode with various functionalized cathodes can create novel dual-mode responsive formaldehyde sensors. Dong and Zhai et al. introduced a novel self-powered biosensor (ESPB) for formaldehyde detection [39]. The key highlights and innovations of this sensor include the unique combination of components, including FDH/poly(methylene green) (PMG)/primary buckypaper (BP) as the bioanode and ferricyanide blue (PB)/gold nanoparticles (Au NPs)/carbon fiber paper (CFP) as the cathode (Figure 4). This innovative design enables self-powered formaldehyde detection without the need for an external power source. During operation, ESPB relies on the enzyme-catalyzed oxidation of formaldehyde occurring at the bioanode, generating NADH as fuel for the electrochemical reaction, thereby powering the device. Additionally, ESPB offers two formaldehyde detection modes: it can directly measure changes in short-circuit current and observe changes in cathode color, enhancing the sensor's reliability. ESPB exhibits excellent detection performance. In the electrochemical detection mode, it boasts a linear detection range from 0.01 to 0.35 mM and a low detection limit of as little as 0.006 mM, with sensitivity comparable to or better than existing formaldehyde sensors. In the colorimetric response mode, the detection response range is 0.010 to 0.045 mM. Furthermore, ESPB demonstrates high selectivity for formaldehyde, with minimal interference from common contaminants such as acetaldehyde and ethanol. It also displays robust long-term stability and has been successfully applied to the detection of formaldehyde in real samples, including tap water and lake water. Building upon these foundations, the research team developed an electrochemical sensor capable of detecting gaseous formaldehyde [17]. The sensor integrates a bioanode and a colorimetric-responsive cathode on a planar indium tin oxide (ITO) conductive substrate. The planar structure of the sensor is covered with a poly(vinyl alcohol) (PVA) gel electrolyte, which provides sufficient internal lateral resistance. This resistance allows the patterned Prussian blue (PB) cathode to gradually change color when exposed to gaseous formaldehyde. The sensor exhibits significant responses to gaseous formaldehyde over a wide concentration range, ranging from 80 to 3000 ppb. Furthermore, the sensor's ability to detect real-world formaldehyde emissions from plywood was practically evaluated, confirming its practicality and reliability in real-world scenarios. This research offers a rapid, dependable, and portable tool for on-site measurement of indoor gaseous formaldehyde levels. Its attractive features include a straightforward setup, ease of operation, and the capacity to operate without external power sources, making it a valuable instrument for assessing indoor air quality and delivering timely exposure risk warnings.

Copper exhibits a rich array of oxidation states, enabling various redox reactions to occur on the electrode's surface, yielding different intermediates that facilitate electron exchange with other redox species. In their study, Song, Guo, and Liang et al. introduced an innovative and highly sensitive formaldehyde detection method based on the bioelectrocatalytic properties of ALDH (aldehyde dehydrogenases) and a copper electrode [35]. The reaction process is depicted in Figure 5 and described by Equations (4)–(6). During the cyclic voltammetry anodic scan, the Cu electrode sequentially generates $Cu_2O$, Cu(II), and Cu(III) oxide and hydroxide combinations. Upon the reverse scan (cathodic process), high-valence copper species are reduced, leading to the appearance of a cathodic peak around −0.2 V corresponding to the Cu(II)/Cu(I) transition. In the presence of formaldehyde and ALDH, an increased amount of Cu(II) is generated during the anodic scan (Equation (7)). Consequently, the cathodic peak at −0.2 V significantly increases. Based on this principle, high-sensitivity formaldehyde detection can be achieved. This research harnesses the catalytic prowess of ALDH to selectively oxidize formaldehyde on the copper electrode. This enzymatic approach ensures a high degree of specificity for formaldehyde. The detection signal demonstrates an extensive linear range spanning from $10^{-15}$ M to $10^{-5}$ M, effectively covering the indoor safe exposure limit for formaldehyde, which stands at approximately $10^{-9}$ M. Impressively, this method boasts an exceptional LOD of $1.46 \times 10^{-15}$ M. When

juxtaposed with traditional detection methods, this approach offers notable advantages in terms of specificity, sensitivity, and stability.

$$2Cu^0 + 2OH^- \rightleftharpoons Cu_2^IO + H_2O + 2e^- \tag{4}$$

$$Cu_2^IO + 2OH^- + H_2O \rightleftharpoons 2Cu^{II}(OH)_2 + 2e^- \tag{5}$$

$$Cu^{II}(OH)_2 + OH^- \rightleftharpoons 2Cu^{III}OOH + H_2O + e^- \tag{6}$$

$$2Cu^{III}OOH + HCHO + H_2O \xrightarrow{ALDH} 2Cu^{II}(OH)_2 + HCOOH \tag{7}$$

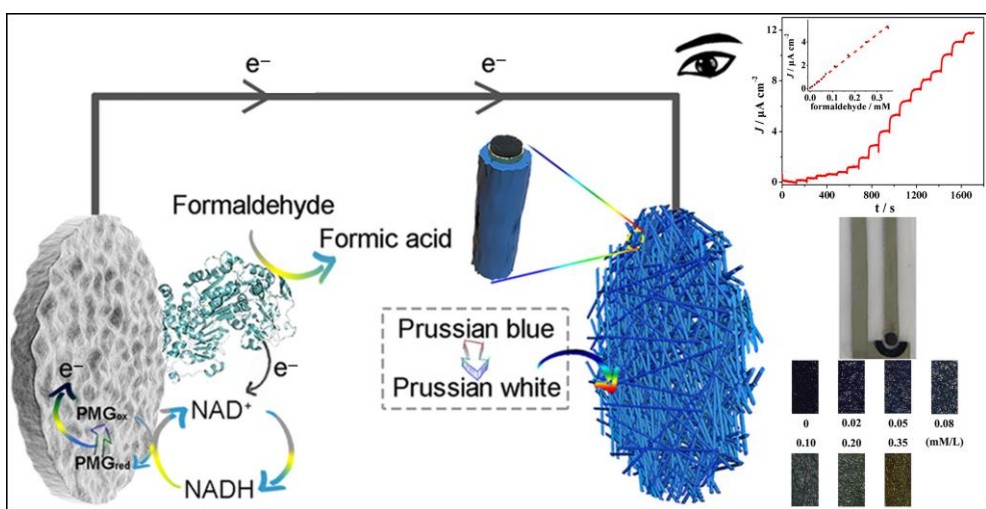

**Figure 4.** Schematic representation of the self-powered biosensor for formaldehyde detection. The illustration depicts the anodic oxidation current response and the colorimetric response based on cathodic Prussian blue color change. Reproduced with permission [39]. Elsevier. Copyright 2019 American Chemical Society.

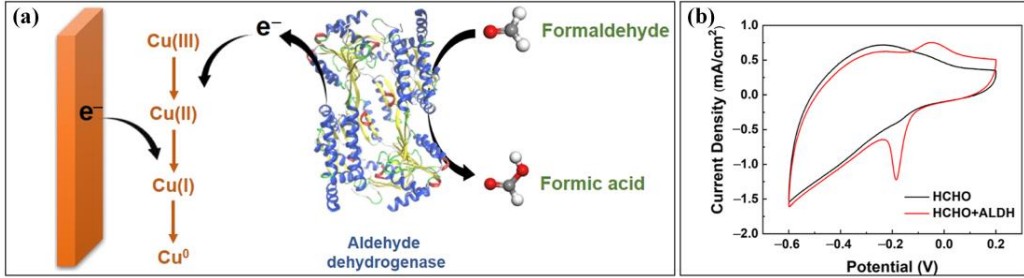

**Figure 5.** (**a**) Schematic diagram of the ALDH-catalyzed oxidation process of formaldehyde on the Cu electrode surface. (**b**) Cyclic voltammetry scans of the Cu electrode in formaldehyde solution with the presence and absence of ALDH. Reproduced with permission [35]. Elsevier. Copyright 2022 American Chemical Society.

### 3.1.2. Alcohol-Oxidase-Based Formaldehyde Sensors

Alcohol oxidase (EC 1.1.3.13) is a crucial enzyme that catalyzes the oxidation of primary alcohols, including methanol. Studies have revealed that AOX can also catalyze the oxidation of formaldehyde to produce formic acid. AOX itself undergoes reduction as it accepts electrons from both methanol and formaldehyde. The reduced form of AOX is subsequently regenerated through oxidation by molecular oxygen ($O_2$), leading to the

generation of hydrogen peroxide ($H_2O_2$) as a byproduct. The overall reaction can be summarized as follows:

$$HCHO + O_2 + H_2O \overset{AOX}{\rightarrow} HCOOH + H_2O_2 \tag{8}$$

Therefore, AOX can be employed to fabricate electrochemical sensors for formaldehyde detection. Depending on the products and processes involved in the catalytic reaction, various types of electrochemical sensors can be developed. For instance, since formic acid can release hydrogen ions, a potential-based sensor with ion-selective membranes for hydrogen ion recognition can be constructed [78,81]. Alternatively, conductivity-based sensors can be developed by leveraging the increase in solution conductivity caused by the presence of ions [77]. Furthermore, the oxidative–reductive behavior associated with the product hydrogen peroxide can be utilized to create current-based formaldehyde sensors [79]. Additionally, as this reaction depletes oxygen from the solution, quantitative formaldehyde analysis can also be accomplished using an oxygen electrode [82].

Korpan et al. were the first to develop highly selective and stable formaldehyde sensors based on pH-sensitive field-effect transistors [83]. The study also discusses the reasons for the high selectivity of these sensors, suggesting that the immobilization of AOX on the transistor's surface is a key factor. This immobilization likely enhances selectivity by promoting the binding of formaldehyde to the amino groups on the enzyme's surface, facilitating preferential adsorption and reaction, thereby improving detection selectivity. Subsequently, Heng et al. developed a formaldehyde biosensor that utilizes hydrophobic acrylic acid microspheres to immobilize AOX [78]. These microspheres were synthesized through photopolymerization and deposited onto a pH sensor equipped with a polypropylene membrane and a Ag/AgCl electrode. AOX catalyzes the oxidation of formaldehyde, producing protons that can be measured using the pH sensor. This biosensor exhibited a linear response to formaldehyde concentrations ranging from 0.3 to 316.2 mM, with a detection limit of 0.3 mM. The study emphasizes that compared to thick-film methods, acrylic acid microspheres improve the biosensor's response time, linear response range, and stability.

Recently, Nurlely et al. have developed an efficient, stable, and highly sensitive formaldehyde sensor [81]. The electrode preparation process involves first photo-induced polymerization of poly(2-hydroxyethyl methacrylate) (pHEMA) on a Ag/AgCl screen-printed electrode, followed by the photopolymerization of a pnBA-NAS $H^+$ ion-selective membrane on its surface. Subsequently, the AOX enzyme is covalently immobilized on the membrane's surface via the spontaneously formed amide bond between the succinimide functional group of the acrylic membrane and the amine functional group of the enzyme. The working principle is based on covalently anchoring the AOX enzyme on the membrane's surface, allowing formaldehyde to undergo a catalytic reaction with the enzyme, resulting in a change in the interface pH, thereby inducing a potential difference. The potential difference is measured using a silver/silver chloride (Ag/AgCl) electrode as an indicator of formaldehyde concentration (Figure 6). The sensor exhibits a linear response to formaldehyde concentrations ranging from 0.5 to 220.0 mM, with high sensitivity (59.23 ± 0.85 mV/decade) and a low detection limit (0.1 mM). Furthermore, the sensor demonstrates a rapid response time, extended operational lifespan, and excellent reproducibility and repeatability. Validation against the standard Nash method confirms the reliability of the sensor in detecting formaldehyde concentrations in commercial fish samples, offering a promising approach for formaldehyde detection in the field of food safety.

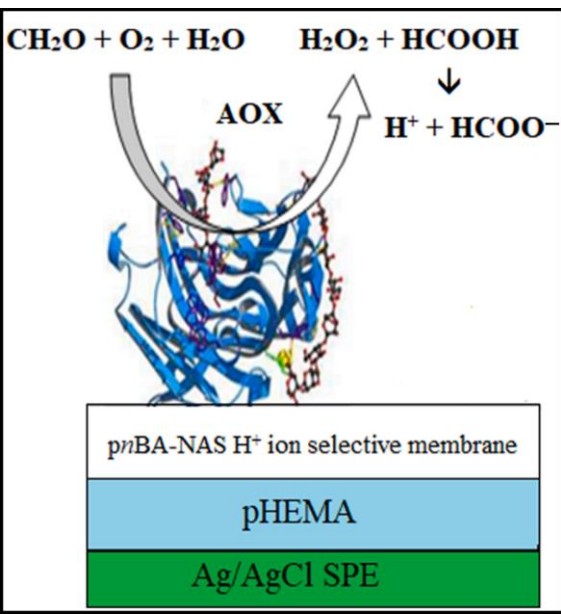

**Figure 6.** Schematic representation of the electrode structure in the potentiometric biosensor for formaldehyde detection. Reproduced with permission [81]. Copyright 2021 Elsevier.

*3.2. Electrochemical Sensors Rely on Electrocatalysts*

Using various electrocatalysts for the electrochemical oxidation of formaldehyde to construct electrochemical sensors for its detection in solution has become a hot research topic, especially in the last decade, with dozens of related reports emerging. The electrocatalysts employed cover a wide range, including metal catalysts such as Ag, Au, Cu, Ni, Pd, Pt, etc.; bimetallic catalysts like AgPd, CrPd, CuPd, PdPt, PtAg, PtPd, PtRu, etc.; metal oxides or hydroxides such as $CeO_2$, $Co(OH)_2$, $CuO$, $MnO_2$, $MoOx$, $Ni(OH)_2$, $NiO$, $SnO_2/NiO$, $TiO_2/RuO_2$, $ZnO$, etc.; as well as non-metallic elemental materials, including various carbon materials and polymer materials like MiPAN@GP (molecular imprinted polymer of acrylonitrile@graphite paste), pDA (polydopamine), polypyrrole/graphene, etc., and composite materials composed of these components. Below, we will discuss them in categories (Table 2).

3.2.1. Noble-Metal-Based Formaldehyde Sensors

Noble metal catalysts are important electrocatalysts, widely utilized in various fields such as oxygen reduction, hydrogen evolution, oxygen evolution, $CO_2$ electroreduction, and oxidation of organic small molecules. Precious metal catalysts, including gold (Au), platinum (Pt), palladium (Pd), and silver (Ag), have also proven highly effective in the electrocatalytic oxidation of formaldehyde. They can efficiently and directly convert formaldehyde into carbon dioxide and water through different pathways, making them essential materials for constructing electrochemical formaldehyde sensors. Furthermore, their surface characteristics can be adjusted to optimize reaction pathways and enhance efficiency. Current research focuses on manipulating nanostructures and selecting suitable support substrates to improve the performance of these catalysts, aiming for better catalytic activity and stability.

- Au-based electrocatalysts

In as early as the year 2000, Hauser et al. devised a method for the direct amperometric detection of low concentrations of formaldehyde in the gas phase [84]. This method involved depositing gold onto a Nafion membrane as the working electrode. The study also explored the impact of gas flow rate and gas stream humidity on the sensor's performance, along with its sensitivity to various organic and inorganic gases. Furthermore, the researchers proposed a solution to counter interference from NO, $NO_2$, and $SO_2$ by

selectively adsorbing formaldehyde from the sample stream using an aluminum oxide filter. This approach boasts a low detection limit and an extensive dynamic range, making it highly suitable for continuous air monitoring applications. Subsequently, Surareungchai et al. achieved the sensing of formaldehyde using an unmodified gold electrode through the application of pulsed amperometric detection (PAD) in a flow injection (FI) system [85]. Mendoza et al. developed an electrochemical sensor for detecting formaldehyde in water samples by employing gold nanoclusters (Au NCs) to modify an SPCE (screen-printed carbon electrode) [86]. This sensor is highly resistant to interference from glucose, formic acid, methanol, or ethanol, making it well suited for practical applications.

In recent years, there have been studies employing various methods to prepare nanostructured gold materials with precise structural features, aiming to enhance their catalytic performance. Kumar et al. utilized an in situ electrochemical process to fabricate gold (111)-oriented nanoparticles on the surface of carbon nanofibers–chitosan composite (GCE/CNF-CHIT@Au$_{nano}$) (Figure 7a) [87]. This innovative approach resulted in the development of a sensor capable of detecting formaldehyde in buffered solutions. Through a range of physicochemical characterization techniques, it was revealed that amino-functionalized chitosan stabilizes gold (111) nanoparticles with a size of approximately 10 nm within the composite matrix. In comparison to other catalysts, these Au(111) catalysts exhibit exceptional catalytic activity, stability, and remarkable surface characteristics. The mechanism of formaldehyde electrocatalytic oxidation on the GCE/CNF-CHIT@Au$_{nano}$ modified electrode was investigated using cyclic voltammetry (CV) and electrochemical quartz crystal microbalance (EQCM). During the anodic scan, formaldehyde exhibited oxidation onset at 0.15 V, followed by a broad anodic peak at 0.4 V. In the reverse scan, a well-defined anodic peak was observed at the same potential. The mechanistic pathway for formaldehyde oxidation involves two possible routes: a direct pathway (dehydrogenation) and an indirect pathway (dehydration) via the adsorption of CO as a poisoning species (Figure 7b). The preferred direct pathway leads to the formation of formate-type intermediates, ultimately oxidizing to $CO_2$. On the other hand, the indirect pathway involves the production of CO, which further oxidizes to $CO_2$. The direct pathway is favored due to its avoidance of CO formation, which can obstruct active surface sites. Notably, in this assay, a specific molecular mass of $44 \pm 1$ g·mol$^{-1}$ corresponding to formate formation was detected, providing confirmation of formate as an intermediate species in the overall reaction. These findings suggest that the sensor primarily follows the direct pathway, involving the formation of formate intermediates, leading to the production of $CO_2$. This study sheds light on the electrocatalytic behavior of the modified electrode and the role of specific intermediates in the reaction mechanism. Furthermore, the sensor's applicability for detecting formaldehyde in commercial hair dye products was demonstrated, achieving an impressive recovery rate of approximately 100%.

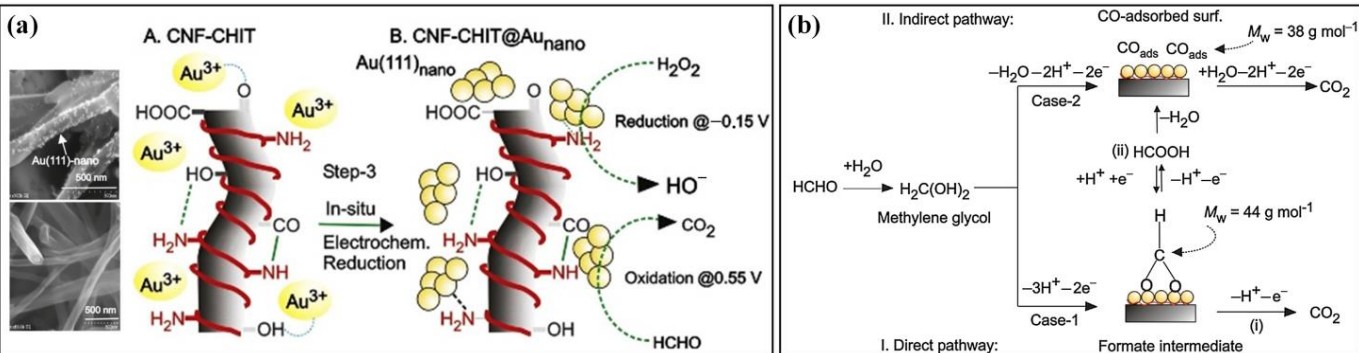

**Figure 7.** (**a**) Schematic depiction of Au(111) nanoparticle synthesis and SEM characterization results; (**b**) oxidation pathway of formaldehyde on the electrocatalyst surface. Reproduced with permission [87]. Copyright 2017 Elsevier.

**Table 2.** Electrochemical sensors rely on electrocatalysts.

| Electrocatalysts | Electrode Materials | Electrolyte | Dynamic Range | Detection Limit | Applications | Ref. |
|---|---|---|---|---|---|---|
| Au | Nafion/Au | $H_2SO_4$ | 13 ppb–10 ppm | 13 ppb | | [84] |
| Au | Au disk electrode | NaOH | 0–10 mM | 0.0129 mM | | [85] |
| Au NCs | SPCE/Au NCs | NaOH | 1–10 mM | 0.93 mM | Water samples | [86] |
| Au NPs | GCE/CNF-CHIT@Au$_{nano}$ | Phosphate buffer | 0.1–10 mM | 3.54 $\mu g \cdot L^{-1}$ | Hair dye | [87] |
| Au NPs | AuNPs/PPy/GCE | NaOH | 0.4–2.1 mM | 0.4 mM | Milk | [88] |
| Au NCs | PDA@Au NCs-MIPs/HOPG | $H_2SO_4$ | 0.2 $\mu$M–0.02 M | 0.1 $\mu$M | Octopuses | [89] |
| Au | Au electrode | IL/DMSO | 5.30–53.00 $\mu$M | 0.53$\mu$M | | [90] |
| Pt NPs | WGE/Ch/MWCNTs/PAN/Pt | $H_2SO_4$ | $10^{-9}$–$10^{-3}$ M | $4.6 \times 10^{-11}$ M | | [91] |
| Pt NPs | Pt NPs-SPUME | Nafion films | 0–5.1 ppm | 80 ppb | Gas | [92] |
| Pt NPs | GCE/Pt NPs/L-alanine | NaOH | 0.3–1050 $\mu$mol/L | 0.14 $\mu$M | Water | [93] |
| Pt | GCE/Graphene–Pt | $H_2SO_4$ | 0–2 mM | 0.04 mM | | [94] |
| Pt | SPE/Pt | NaOH | 100–1000 $\mu$mol/L | | Vegetable | [95] |
| Pd NPs | Ti/Pd NPs | NaOH | 0-20 mM | 38.6 $\mu$M | | [96] |
| Pd NWs | GCE/Pd NWs | NaOH | 2 $\mu$M to 1 mM | 0.5 $\mu$M | | [97] |
| Pd | GCE/Nafion-Graphene-Pd | NaOH | 7.75–62.0 $\mu$M | 3.15$\mu$M | | [98] |
| Pd NPs | GCE/Pd NPs | NaOH | 30 $\mu$M–14 mM | 10$\mu$M | | [99] |
| Pd NPs | GCE/GO-BDMA-Pd | KOH | 1 $\mu$M–18 mM | 0.35$\mu$M | Tomato sauce | [100] |
| Pd NPs | GCE/GO-PAA-Pd NPs | KOH | 50 $\mu$M–50 mM | 16 $\mu$M | Food | [101] |
| Pd | GCE/Ppy-Pd | NaOH | 0.001–0.1 mM | 0.9$\mu$M | | [102] |
| Pd | GCE/Nafion@rGO-Pd | NaOH | 2–20 mM | | | [103] |
| Pd NPs | GCE/CMs-Pd NPs | NaOH | 0.025–15.00 mM | 8 $\mu$M | Seafood | [104] |
| Pdnano | nanoPd@LIG | KOH | 0.01–4.0 mM | 6.4 $\mu$M | Seafood, etc. | [105] |
| Ag NPs | GCE/GOPx-Ag NPs | KOH | 1 $\mu$M–70 mM | 0.167 $\mu$M | | [106] |
| Ag Nanoporous | Cu/Ag | KOH | 10–100 mM | | | [107] |
| Ag Nanoporous | Cu/Ag | KOH | 10–100 mM | | | [108] |
| Pt-Ru | PtRu black anode | Nafion®-117 | 0.002–1.25 $g \cdot L^{-1}$ | -- | Gas | [109] |
| Pt-Pd | GCE/Nf/Pt–Pd | $H_2SO_4$ | 10 $\mu$M–1 mM | 3 $\mu$M | Water | [110] |
| Pd-Au | CILE/Au/Pd | NaOH | 0.015–1.4 mM 1.4-56.7 mM | 3 $\mu$M | | [111] |
| Ag-Pd | CILE/AgPd | NaOH | 10 $\mu$M–70 mM | 2 $\mu$M | Water | [112] |
| Sn-Pt | Ti/Sn/nanoPt | $H_2SO_4$ | 0.003–0.1 M | 0.506 mM | | [113] |
| Cu-Pd | Glass slides/Cu-Pd | NaOH | -- | -- | -- | [114] |
| Pd-Pt | GCE/Nafion-graphene-Pd-Pt | $H_2SO_4$ | 4.50 $\mu$M–0.180 mM | 2.85$\mu$M | Water | [115] |
| Pd-Cu | Pd-Cu-SBA-16/CPE | NaOH | 1.79–121.86 mM | 16 $\mu$M | -- | [116] |
| Ni-Pd | Ni-Pd/GCE | NaOH | 10 $\mu$M–1 mM | 5.4$\mu$M | Water | [117] |
| Pt–Ag | Pt–Ag/rGO/SPE | | 1–100 $\mu$M | 1 $\mu$M | Juice, etc. | [118] |
| Cr-Pdene | Cr-doped Pd metallene (Cr-Pdene)/GCE | | 1–5$\mu$M | 1 $\mu$M | Gas, Food | [119] |
| Ni | BPE-Ni | NaOH | 0.037–10 $\mu g \cdot mL^{-1}$ | 0.23 $\mu g \cdot L^{-1}$ | -- | [120] |
| Ni(OH)$_2$ | Ni/Ni(OH)$_2$ | NaOH | 70 $\mu$M–16 mM | 20 $\mu$M | -- | [121] |
| Ni(OH)$_2$ | CPE/Ni/P(Ni-doped P nanozeolite) | NaOH | 0.02–11.5 mM | 5.8 $\mu$M | Water | [122] |
| Ni(OH)$_2$ | TNAs/Ni/Ni(OH)$_2$ | KOH | 1.3–13 mM | 33.4 $\mu$M | | [123] |
| Ni(OH)$_2$ | CPE/NaA nanozeolite/Ni(OH)$_2$ | NaOH | 6.0–231 $\mu$M | 2 $\mu$M | Water | [124] |
| Ni(OH)$_2$ | GCE/Ni-Ni(OH)$_2$ | KOH | 0.01–1 mM | 6 $\mu$M | Water | [125] |
| NiWO$_4$ | CPE/NiWO$_4$ | NaOH | 0.008–1 mM | 3.6 $\mu$M | Water | [126] |
| Ni(OH)$_2$ | Ni-NWs/Ni(OH)$_2$ | KOH | 20 $\mu$M–2 mM 2–20 mM | 0.8 $\mu$M | -- | [127] |



**Table 2.** *Cont.*

| Electrocatalysts | Electrode Materials | Electrolyte | Dynamic Range | Detection Limit | Applications | Ref. |
|---|---|---|---|---|---|---|
| $Ni(OH)_2$ | $FTO/Ni/Ni(OH)_2$ | KOH | 0–6.5 mM | 8.3 μM | Juice | [128] |
| NiO | CC (carbon cloth)/NiO | NaOH | 5 μM–3 mM | 7.45 nM | Water | [129] |
| $Ni(OH)_2$ | $GCE/rGO/Ni(OH)_2$ | KOH | 0.1–100 mM | 60 μM | Wastewater | [130] |
| $Ni_3S_2$ | Nickel foam (NF)/$Ni_3S_2$ | NaOH | 0.002–5.45 mM | 1.23 μM | Water | [131] |
| $SnO_2@NiO$ | $GCE/SnO_2@NiO$ | NaOH | 0.1–28 mM | 2.8 nM | -- | [132] |
| $Ni/Ni(OH)_2$ | $CNFs/Ni/Ni(OH)_2$ | NaOH | 0.05–91.5 mM, 0.5–10 ppm | 0.36 ppm | Gas | [133] |
| $Ni(OH)_2$ | $SPEs/Ni NWs/Ni(OH)_2$ | NaOH | 0.8 μM–10 mM | LOQ: 0.8 μM | Water | [134] |
| Cu | Copper Electrode | NaOH | 1 μM–1 mM | 0.019 mM | -- | [135] |
| CuO NPs | GCE/CuO NPs | NaOH | 1.0 μM–10.0 mM | 0.25 μM | -- | [136] |
| Cu–Ni | Cu–Ni | NaOH | 3–100 mM | -- | -- | [137] |
| $Cu_2O/CuO$ | $CILE/Cu_2O/CuO$ | NaOH | 0.1–110 mM | 10 μM | -- | [138] |
| Cu NPs | SPCE/PS/Cu NPs | NaOH | 0.4–4 mM 1–300 mM | 0.0124 μM | Water | [139] |
| CuO/ZnO | GCE/CZO(Cu-codoped ZnO) | Phosphate buffer | 2 nM–6 mM | 4.1 nM | Water | [140] |
| CuO-CuOOH | $Ti/TiO_2/Cu/CuO$ | NaOH | 65 μM–7.8 mM | 25.0 μM | -- | [141] |
| CuO | CC/CuO | NaOH | 20 μM–3 mM | 26 nM | Milk | [142] |
| CuO | CuO/polyaniline (PANI) | Phosphate buffer | -- | 1 μM | -- | [143] |
| $TiO_2/RuO_2$ | $Ti/Ru_{0.3}Ti_{0.7}O_2$ | K2SO4 | -- | -- | -- | [144] |
| $MnO_2$ | $GCE/MnO_2$ | Na2SO4/H2SO4 | 0.02–0.2 mM 0.2–2 mM | 10.2 μM | -- | [145] |
| $MoO_x$ | SPGE/Carbon/$MoO_x$-Nafion | Nafion | -- | 60 ppb | Gas | [146] |
| $Co(OH)_2$ | $CC/Co(OH)_2$ nanosheet arrays | NaOH | 4 μM–5.45 mM | 0.57 μM | Not specified | [147] |
| $Ag_2S@g\text{-}C_3N_4$ | $GCE/NiFe_2O_4/Ag_2S@g\text{-}C_3N_4$ | Phosphate buffer | 0.9–120 mM | 1.63 μM | -- | [148] |
| $GP@CeO_2$ | $GP@CeO_2$ | NaOH | 25–120 μM 120–1000 μM | 1 μM | Mushroom | [149] |
| ZnO | SPCE/Egg albumin@ZnO SPCE/Chitosan@ZnO | Phosphate buffer | 1–5 μM 1–9 μM | 6.2 nM | Urine | [150] |
| ZnO | GPE/ZnO NPs | Phosphate buffer | 0–100 mM | 18 μM | -- | [151] |
| Polypyrrole | GCE/Graphene@Polypyrrole | KCl | 0.001–2 mM | 0.028 μM | -- | [152] |
| pDA | Stainless Steel Electrode/pDA | $H_2SO_4$ | 0.43–1.60 μM | 0.14 μM | Fish | [153] |
| MiPAN@GP | GP@MiPAN | NaOH | 10 μM–1 mM | 0.63 μM | Mushroom, fish | [154] |
| PANI/GO | GE/PANI@GO | $HClO_4$ | 0.1–20 μM | 0.0185 μM | -- | [155] |

Deposition of gold nanoparticles onto polymer surfaces has proven to be an effective technique, allowing for the efficient binding and dispersion of nanoparticles due to the functional groups present on the polymer surface. Huang et al. were the first to modify a glassy carbon electrode (GCE) with polypyrrole (PPy) and subsequently electrodeposit gold nanoparticles (AuNPs), resulting in a functional electrode denoted as GCE/PPy/AuNPs [88]. This electrode exhibited high electrical conductivity and excellent catalytic activity, enabling the sensitive quantification of formaldehyde. Liu et al. presented the development of a molecularly imprinted electrochemical sensing platform tailored for the specific detection of formaldehyde [89]. They immobilized gold nanoclusters (Au NCs) onto the surface of polydopamine nanospheres (pDA NPs) to create a pDA@Au NCs composite material. Subsequently, molecularly imprinted polymers (MIPs) for formaldehyde were synthesized on the surface of pDA@Au NCs, yielding the pDA@Au NCs-MIPs-HOPG (highly oriented pyrolytic graphite) electrode. This sensor, combining molecular imprinting technology (MIT) with the catalytic properties of noble metal nanoparticles (Au NCs), demonstrated impressive selectivity and high sensitivity. It featured a broad detection range spanning from 0.2 μM to 0.02 M and a low detection limit of 0.1 μM. The sensor was effectively applied to detect trace amounts of formaldehyde residues in seafood, par-

ticularly octopuses, showcasing satisfactory selectivity and reproducibility. Li and Dong et al. reported an electrochemical method based on ionic liquid (IL) electrolytes for the species-selective detection of volatile organic compounds, including formaldehyde [90]. This approach relies on the distinct electrochemical behaviors of different target molecules on a gold electrode surface in the presence of IL electrolyte and utilizes linear discriminant analysis (LDA) to achieve species-selective detection.

- Pt-based electrocatalysts

Similar to gold nanoparticles, platinum nanoparticles also possess high specific surface area, excellent electrocatalytic activity, and stability. They have significant potential for use as electrocatalysts in constructing formaldehyde electrochemical sensors. Platinum nanoparticles can promote the oxidation reaction of formaldehyde, thereby enhancing the current response and enabling rapid and accurate detection of formaldehyde concentrations. Furthermore, the nanoscale size and tunable structure of these materials provide ample room for further optimizing sensor performance, including improving selectivity and reducing interference. Peng et al. were the first to develop an electrochemical sensor for formaldehyde detection based on platinum nanoparticles [91]. The sensor consisted of platinum nanoparticles deposited on multi-walled carbon nanotubes (MWCNTs) coated with polyaniline (PAN). This configuration enhanced the electrocatalytic oxidation of formaldehyde. The sensor exhibited improved analytical performance, demonstrating high sensitivity and selectivity for formaldehyde detection in various test environments.

Zen et al. developed an innovative electrochemical sensor for formaldehyde gas [92], featuring a platinum-coated screen-printed ultramicroelectrode wrapped in Nafion as the electrolyte. This sensor's novelty stems from its detection mechanism, which uses high oxidation potential to transform formaldehyde into formic acid, thereby activating Pt catalyst sites. Employing square wave voltammetry, the sensor distinctly separates responses from platinum oxide reduction and formic acid oxidation, achieving a broad linear detection range with an impressive sensitivity down to 80 ppb. In a related study, Cai and Song crafted a platinum nanoparticle (Pt-NP) and L-alanine-modified electrode through electrodeposition and self-assembly [93]. Their investigations into various factors affecting formaldehyde's electrocatalytic oxidation revealed a highly responsive sensor with a detection limit of 0.14 μM, highlighting the synergistic benefits of combining Pt nanoparticles with L-alanine.

Gao et al. developed a sensitive electrochemical sensor for formaldehyde detection using a composite electrode composed of directly electrodeposited graphene and platinum nanoparticles [94]. This electrode design combined graphene's high surface area and conductivity with platinum's excellent catalytic capability. The sensor relies on the electrocatalytic oxidation of formaldehyde at the electrode surface, resulting in enhanced detection sensitivity. It offers a broad linear range and a low detection limit for formaldehyde, along with good reproducibility and stability, making it effective for formaldehyde analysis in various environments. Wu and Hua et al. presented a practical and efficient pocket-sized device for detecting formaldehyde adulteration in vegetables [95]. This device integrates a low-cost, handheld detector with an SPE amperometric sensor and a potentiostat. It can detect formaldehyde concentrations as low as 100 μmol·L$^{-1}$ and is effective in a range between 100 and 1000 μmol·L$^{-1}$. This sensor was tested on 53 vegetable samples, successfully identifying formaldehyde contamination.

- Pd-based electrocatalysts

Palladium (Pd) nanomaterials exhibit high catalytic activity in the electro-oxidation of organic small molecules. Furthermore, compared to Pt, Pd-containing materials are particularly attractive due to their relatively lower cost and high tolerance to CO formation as a byproduct during formaldehyde oxidation. The use of nanoparticles with high electroactive surface areas can enhance the sensitivity of these materials to formaldehyde oxidation. Yi et al. employed a hydrothermal method, using PdCl$_2$, EDTA (ethylenediaminetetraacetic acid), and formaldehyde as precursors, to prepare unique three-dimensional porous Pd

nanoparticles on a titanium (Ti) substrate [96]. These nanostructured Pd electrodes exhibited outstanding electrocatalytic performance for formaldehyde oxidation in alkaline solutions. They demonstrated a low onset potential for formaldehyde electro-oxidation on the nanoPd electrode, approximately $-0.85$ V vs. the saturated calomel electrode (SCE), along with a significantly large anodic current density of 66.96 mA·cm$^2$. Cheng et al. successfully fabricated palladium nanowire arrays (Pd NW arrays) on a glassy carbon electrode surface using an anodized aluminum oxide template electrodeposition method, resulting in a unique electrode structure [97]. Their research revealed that the electro-oxidation of formaldehyde on the Pd NW array electrode effectively suppressed the formation of the toxic intermediate CO.

Dong et al. have developed a novel electrochemical sensor for the detection of formaldehyde using palladium–graphene nanohybrids [98]. Initially, Pd–graphene nanohybrids were synthesized via a straightforward chemical reduction method. These Pd–graphene nanohybrids were then dispersed in a Nafion solution and employed to modify a glassy carbon electrode. This modified Pd–graphene–Nafion/GCE electrode displayed exceptional electrocatalytic activity for the oxidation of formaldehyde in an alkaline medium. The observed peak current exhibited a linear correlation with the formaldehyde concentration within the range of 7.75 μM to 62.0 μM, with a detection limit of 3.15 μM. Furthermore, palladium nanoparticles (Pd NPs) were prepared on a glassy carbon electrode using cyclic voltammetry (CV) and potentiostatic techniques for the electrocatalytic oxidation of formaldehyde [99].

Jeon et al. synthesized GO-BDMA-Pd nanocomposite material using graphene functionalized with 1,4-benzenedimethaneamine (BDMA) via a material chemical reduction method [100]. TEM images demonstrated excellent dispersion of palladium nanoparticles (Pd NPs) on the surface of GO-BDMA. The GO-BDMA-Pd sensor exhibited high sensitivity, good stability, rapid response, and a wide linear range of $1 \times 10^{-6}$ M to $1.8 \times 10^{-2}$ M, with a low LOD ($3.5 \times 10^{-7}$ M). Furthermore, the study revealed that the GO-BDMA-Pd sensor inhibited the formation of toxic intermediates, such as CO, during the electro-oxidation of formaldehyde. Limbut et al. employed a laser irradiation process to fabricate nano-palladium-grafted laser-induced graphene (nanoPd@LIG) and developed an electrochemical sensor for formaldehyde [105]. This sensor was integrated into a smart electrochemical sensing device, enabling on-site quantitative analysis through near-field communication (NFC) with a smartphone. The proposed system was successfully tested with real food samples, specifically mushrooms, demonstrating good correlation with results obtained from a commercial potentiostat and spectrophotometric analysis. In addition, other materials such as PAA (polyacrylic acid) [101], PPy [102], and carbon microspheres [103] have also been employed for the dispersion and stabilization of palladium nanoparticles (Pd NPs) in the construction of electrochemical sensors for catalyzing formaldehyde oxidation. The electrochemical reactions on the electrode surface typically involve palladium in the following reactions (Equations (9)–(11)). The oxidation of the target molecule, formaldehyde (Equation (12)), is closely related to the catalytic activity of Pd-O.

$$\text{Pd} + \text{OH}^- \rightarrow \text{Pd-OH}_{\text{ads}} + \text{e}^- \tag{9}$$

$$\text{Pd-OH}_{\text{ads}} + \text{OH}^- \rightarrow \text{Pd-O(Pd oxide)} + \text{H}_2\text{O} + \text{e}^- \tag{10}$$

$$\text{Pd-O(Pd oxide)} + \text{H}_2\text{O} + 2\text{e}^- \rightarrow \text{Pd} + \text{OH}^- \tag{11}$$

$$\text{HCHO} + \text{H}_2\text{O} \leftrightarrow \text{H}_2\text{C(OH)}_2 \rightarrow \text{HCOOH} + 2\text{H}^+ + 2\text{e}^- \tag{12}$$

- Ag-based electrocatalysts

Nanoporous silver, owing to its high electrical conductivity, large surface area, high porosity, and cost-effectiveness, finds extensive applications in catalyzing the oxidation

of small organic molecules. The catalytic activity of nanoporous silver catalysts is closely linked to their size, morphology, structure, and the physical environment in which they operate. Jeon et al. reported the synthesis of silver nanoparticles (Ag NPs) on graphene oxide functionalized with p-phenylenediamine (Px), resulting in the formation of GOPx-Ag nanocomposites [106]. These nanocomposites were employed as catalysts for the direct electro-oxidation of formaldehyde, with an onset oxidation potential of $-0.783$ V. Cyclic voltammetry tests revealed oxidation–reduction processes ($Ag \leftrightarrow Ag_2O$) occurring on the electrode surface of Ag. In comparison to electrodes lacking Px, GOPx-Ag-modified electrodes exhibited a 57.3% increase in the oxidation response current for formaldehyde and a 159 mV negative shift in oxidation potential. This indicates that Px enhances the dispersion of Ag NPs, leading to higher electrocatalytic activity. The authors also explained that Px-functionalized graphene oxide provides N lone pair electrons. This functionalized GO is a dense two-dimensional (2D) material with unparalleled electrical conductivity. Moreover, the π-π stacking interaction between the aromatic rings of Px and GO not only provides support for anchoring dispersed nanoparticles but also increases the surface area, enhancing electrical conductivity and strong wettability adhesion. This results in excellent charge transfer reactions at the electrode–electrolyte interface. The sensor exhibited a broad linear detection range for formaldehyde from 1 μM to 70 mM, with a detection limit of 0.167 μM and a sensitivity of 35.74 $\mu A \cdot mM^{-1} \cdot cm^{-2}$. Furthermore, the GOPx-Ag catalyst demonstrated outstanding long-term stability and resistance to poisoning.

Dan et al. utilized two electrochemical dealloying methods to fabricate nanoporous silver with a three-dimensional continuous interconnected structure [107]. Initially, $Ag_{30}Zn_{70}$ precursor alloy with uniform composition was obtained through high-frequency induction melting. Subsequently, the alloy was re-melted using high-frequency induction heating and processed into strip samples for use as working electrodes. Then, zinc was dissolved using the constant potential method to prepare nanoporous silver. Nanoporous silver exhibited superior catalytic and detection performance for formaldehyde.

### 3.2.2. Bimetallic-Based Formaldehyde Sensors

Bimetallic nanocatalysts have demonstrated significant advantages in catalyzing the oxidation of formaldehyde due to their unique properties. These catalysts typically consist of two different metals, leading to synergistic effects that enhance catalytic efficiency, selectivity, and stability. They often exhibit superior activity compared to single-metal catalysts, possibly due to improved electron transfer, increased active sites, and altered reaction pathways. Recent research has focused on optimizing the composition, structure, and size of bimetallic nanoparticles to further enhance their performance in formaldehyde oxidation. These advancements include the development of novel synthesis methods, understanding the nanoscale interaction between the two metals, and exploring various combinations of metals. Currently reported bimetallic nanocatalysts for electrochemical detection of formaldehyde include Pt-Ru [109], Pd–Pt [110,115,156], Pd-Au [111], Ag-Pd [112], Sn-Pt [113], Cu-Pd [114,116], Ni-Pd [117], Pt–Ag [118], and Cr-Pd [119]. Due to Pd's high catalytic activity and low susceptibility to poisoning during anodic oxidation, it is commonly used in alloy materials. Bimetallic catalysts comprising Pd and other noble metals have shown excellent catalytic performance in formaldehyde electro-oxidation. For instance, Kang et al. developed a novel electrochemical sensor for detecting formaldehyde by depositing Pt-Pd alloy nanoparticles on a Nafion-coated glassy carbon electrode [110]. Scanning electron microscopy confirmed the uniform dispersion of bimetallic Pt-Pd nanoparticles within the Nafion film. The modified electrode exhibited significant electrocatalytic activity towards formaldehyde oxidation. The Nafion film on the glassy carbon electrode played a crucial role in promoting the dispersion of Pt-Pd nanoparticles, enhancing catalytic activity, and repelling negatively charged interfering species. The electrochemical sensor showed a linear response to formaldehyde in the range of 10 μM to 1 mM, achieving a low detection limit of 3 μM in an acidic solution. Safavi et al. employed underpotential deposition (UPD) to create a Pd-covered layer on gold nanoparticles electrodeposited onto a carbon

ionic liquid electrode (CILE) [111]. This innovative sensor configuration was utilized for formaldehyde detection in aqueous solutions. The Pd-covered layer on gold nanoparticles provided effective surface area and active sites for formaldehyde oxidation, resulting in a highly sensitive current response. Subsequently, the research team synthesized silver–palladium alloy nanoparticles using ionic liquids and microwave radiation [112]. These nanoparticles exhibited enhanced electrocatalytic activity for formaldehyde oxidation compared to pure Pd or Ag nanoparticles. The AgPd/CILE also demonstrated excellent fouling resistance, essential for long-term stability and usage in electrochemical applications.

Mardared et al. investigated the electrocatalytic performance of various compositions of copper–palladium (CuPd) thin film combinatorial libraries using cyclic voltammetry throughout the entire composition range of CuPd films [114]. They found that Cu-7.5 at.% Pd exhibited the highest electrocatalytic activity for formaldehyde oxidation with a starting potential of $-0.35$ V and a current density of $1.81$ mA·cm$^{-2}$. The observed electrocatalytic enhancement at the optimal composition was attributed to the synergistic effects involving Pd concentration, surface properties, and electron density. Azizi developed a highly sensitive electrochemical sensor for formaldehyde detection using novel bimetallic nanoporous Pd-Cu-SBA-16/CPE (carbon paste electrode) [116]. Bimetallic nanoparticles composed of palladium (Pd) and copper (Cu) were incorporated into SBA-16 using an electrochemical replacement reaction. This approach reduced the use of precious metal (Pd) while improving electrocatalytic performance. The electrochemical properties of the Pd-Cu-SBA-16/CPE formaldehyde oxidation sensor were thoroughly investigated using cyclic voltammetry, current analysis, and chronoamperometry. The sensor exhibited excellent electrocatalytic activity with high current density and low formaldehyde oxidation overpotential. Wang and Li et al. developed an electrochemical method for on-site detection of formaldehyde in food using Pt-Ag core–shell nanoparticles as the electrocatalyst [118].

Metallenes, a cutting-edge topic in the field of materials science, have emerged as a novel class of two-dimensional materials composed of single layers of metal atoms, exhibiting unique physical and chemical properties [157–159]. In the realm of electrocatalysis, metallenes hold tremendous potential owing to their high surface area, excellent electrical conductivity, and distinctive electronic structure [160–162]. These attributes render them promising candidates for various applications, including catalyst supports [163,164], energy storage [165,166], and sensors [167,168]. In a recent study by Li, Jiang, Zhang, and Guo et al., a novel chromium-doped palladium metallene (Cr-Pdene) was introduced as an advanced catalyst for formaldehyde sensing (Figure 8) [119]. Cr-Pdene, synthesized through controlled crystal growth direction of palladium metallene and diffusion of chromium atoms, features a few atomic layers in thickness (Figure 8a). A key step involves the low-temperature (80 °C) decomposition of chromium hexacarbonyl (Cr(CO)$_6$) precursor, which restricts the planar growth of the palladium metallene. This structural design results in uniform distribution of palladium and chromium elements within Cr-Pdene. Characterization of Cr-Pdene was carried out using transmission electron microscopy (TEM) and high-angle annular dark-field scanning transmission electron microscopy (HAADF-STEM). The introduction of chromium atoms was found to effectively optimize the electronic structure of palladium (d-band downshift) as revealed by in situ Fourier-transform infrared spectroscopy (FTIR) and density functional theory (DFT) calculations (Figure 8b,c). This substantial weakening of CO binding on palladium enhances the conversion efficiency of CO to CO$_2$. Cr-Pdene exhibits excellent stability and resistance to poisoning, thus maintaining its performance over extended usage. Cr-Pdene/C sensors demonstrated excellent linear response, low detection limits, and rapid response times for formaldehyde detection. These sensors also exhibited minimal interference and excellent reproducibility, making them powerful tools for applications in environmental monitoring, health assessment, and food safety. Integration of these sensors into wireless sensor networks or portable devices enables accurate and stable monitoring of formaldehyde.

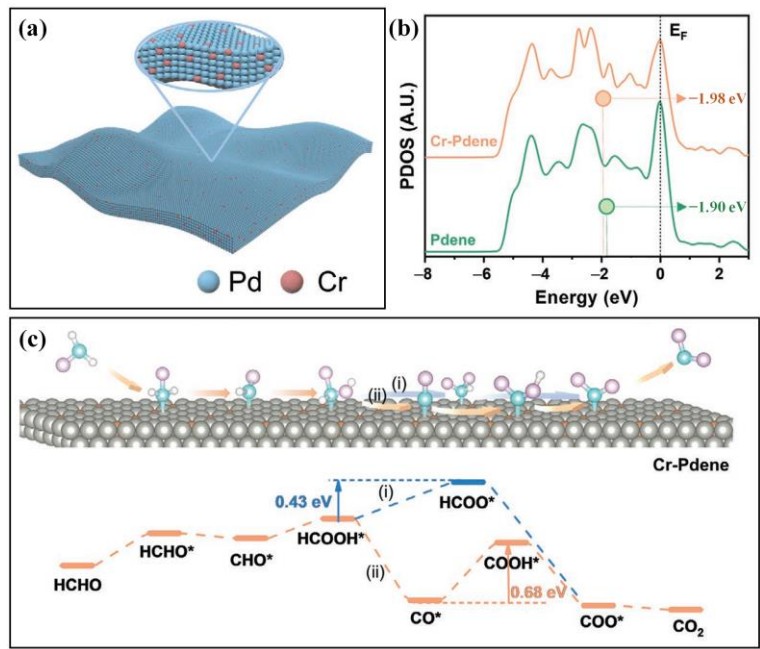

**Figure 8.** (**a**) Schematic model of Cr-Pdene. (**b**) Projected Density of States (PDOS) of the Pd d-band for the Cr-Pdene and Pdene model systems (EF represents the Fermi level). (**c**) Top: schematics illustrating two possible decomposition pathways of intermediates on the Cr-Pdene surface; bottom: free-energy diagrams of intermediates for formaldehyde oxidation on the Cr-Pdene surface; (i) represents the direct pathway (CO-free path), and (ii) represents the indirect pathway (CO-poisoning path). Reproduced with permission [119]. Copyright 2022 Wiley-VCH.

### 3.2.3. Transition Metals and Their Oxide-Based Formaldehyde Sensors

Transition metal materials have shown significant potential in electrochemical catalysis for formaldehyde oxidation. These materials, including nickel, copper, and other metal oxides, are capable of undergoing various oxidation states, which facilitates effective electron mediation required for formaldehyde oxidation. These catalytic materials can be obtained through various preparation methods, such as electrochemical deposition, vapor-phase deposition, chemical corrosion, in situ transformation, and more. They exhibit high catalytic activity for the electrochemical oxidation of formaldehyde and hold great promise in the construction of formaldehyde sensors. The following discussion will categorize these materials and their applications.

- Nickel-based electrocatalysts

Nickel-based transition metal nanomaterials are the most widely used electrocatalysts for constructing electrochemical formaldehyde sensors. They are typically prepared using methods such as electrochemical deposition and chemical corrosion on different substrates. The mechanism of formaldehyde oxidation catalysis by these catalysts is well understood and often involves the conversion of Ni(II) to Ni(III) (Equation (13)). Subsequently, $Ni^{(III)}OOH$ species oxidize the ionized form of formaldehyde, $CH_2(OH)O^-$ (Equations (14) and (15)), and Ni returns to its initial divalent state as $Ni(OH)_2$.

$$Ni^{II}(OH)_2 + OH^- \rightarrow Ni^{III}OOH + H_2O + e^- \tag{13}$$

$$HCHO + H_2O \leftrightarrow CH_2(OH)_2 \overset{OH^-}{\leftrightarrow} CH_2(OH)O^- + H_2O \tag{14}$$

$$Ni^{III}OOH + CH_2(OH)O^- \rightarrow Ni^{II}(OH)_2 + CH_2(O)O^- \tag{15}$$

Ying et al. have introduced a sensitive and cost-effective flow injection analysis (FIA) method for detecting formaldehyde using an activated barrel-plated nickel electrode

(Ni-BPE) [120]. The mechanism of formaldehyde electrocatalytic oxidation on the Ni-BPE electrode in an alkaline medium at ambient temperature is discussed, involving the oxidation of formaldehyde by $Ni^{III}O(OH)$ species. This method exhibits good linearity within the concentration range of 0.037 to 10 μg·mL of formaldehyde, with a low LOD of 0.23 μg·L. The method demonstrates excellent reproducibility and has been successfully applied to the determination of formaldehyde in commercial nail polish samples and drinking water.

Many studies have attempted to develop Ni-based catalysts on different substrate materials. Azizi et al. introduced the development of a nickel-doped P nanozeolite carbon paste electrode (Ni/P-CPE) as an effective sensor for detecting formaldehyde [122]. In this study, a synthetic silica source known as SBA (silica source with boric acid) was used to synthesize P nanozeolite, followed by the incorporation of Ni(II) ions to form Ni(II)-doped P nanozeolite (Ni/P). Nanozeolite, as the substrate, possesses a highly microporous structure and a large surface area, allowing it to provide more active surface area for the even distribution of the catalyst on the electrode surface, thereby enhancing electrocatalytic performance. Furthermore, zeolite materials exhibit molecular sieving properties, selectively adsorbing or allowing the target analyte to diffuse back into the air, thereby improving selectivity in detection. Later on, Hassaninejad-Darzi used synthesized NaA nanozeolite for modifying a carbon paste electrode (CPE) and introduced nickel ions ($Ni^{2+}$) to the electrode, resulting in a $Ni(OH)_2$-NaA/CPE electrode [124]. By loading nickel hydroxide onto the porous structure and large surface area of NaA nanozeolite, the electrode's electrocatalytic activity was enhanced. $TiO_2$, due to its high stability and possible morphologies, is an excellent carrier material that has found widespread applications in electrocatalysis, electrochemical sensors, and other fields [169–175]. Tang et al. prepared $TiO_2$ nanotube arrays (TNAs) using an anodization method and then synthesized $Ni(OH)_2$/Ni nanoparticles in situ on their surfaces through electrochemical deposition and chemical transformation [123]. The resulting $Ni(OH)_2$/Ni/TNA electrode exhibited excellent electrocatalytic activity, rapidly undergoing the oxidation reaction of formaldehyde at a low applied potential (0.35 V) with a fast response time (1 s). Additionally, the rate constant for the oxidation of formaldehyde was as high as $5.36 \times 10^5$ cm$^{-3}$·mol$^{-1}$·s$^{-1}$. Other methods such as aluminum oxide template synthesis [127], electrospinning [132], flame synthesis [129], and more have also been successfully utilized for $Ni(OH)_2$/NiO catalysts with regular nanostructures. Furthermore, other Ni-based catalysts and oxide composites have also shown excellent catalytic performance. Ashkarran et al. investigated an efficient electrochemical platform for the high-performance electro-oxidation of formaldehyde based on a carbon paste electrode modified with amorphous $NiWO_4$ nanoparticles ($NiWO_4$-NPs) [127]. The $NiWO_4$-NPs were synthesized via a simple co-precipitation method. The prepared $NiWO_4$-NPs exhibited a nearly spherical morphology, with tungstate as the major crystalline phase. When the carbon paste electrode (CPE) was modified with $NiWO_4$-NPs, it reduced the overpotential for formaldehyde oxidation and increased the current density compared to the unmodified CPE, as demonstrated by cyclic voltammetry. Jiang and Xiong used a sodium-sulfide-induced chemical etching method with dielectric barrier discharge plasma to synthesize $Ni_3S_2$ nanosheets (NSs) on nickel foam (NF), preparing a $N_{i3}S_2$ NS/NF electrode for the electrochemical detection of formaldehyde in an alkaline solution [131]. The use of sodium sulfide and dielectric barrier discharge plasma on nickel foam allowed for rapid and mild chemical etching, resulting in the formation of $N_{i3}S_2$ nanosheets. This method offers advantages such as speed, mild conditions (room temperature and atmospheric pressure), stability, and reproducibility. The $Ni_3S_2$ NS/NF electrode exhibited excellent electrocatalytic activity for formaldehyde oxidation under alkaline conditions. Sang et al. prepared $SnO_2$-doped NiO heterostructure nanofibers through an electrospinning process and used them to construct an electrochemical sensor for formaldehyde [132]. It was found that NiO nanofibers were oxidized to a weakly conductive p-type semiconductor NiO(OH) in a strong alkaline environment. When pure NiO was doped with a small amount of $SnO_2$ (n-type semiconductor), they formed a PN heterojunction. During the catalytic oxidation of

formaldehyde by NiO(OH), electrons were obtained, increasing the number of electrons in NiO(OH) (p-type). This disrupted the balance of the internal electric field, increased the drift current through the internal electric field, and narrowed the PN junction, making it more conducive to electron conduction (Figure 9). This enhanced the conductivity of NiO(OH). Ultimately, electrochemical tests indicated that SnO2/NiO NFs (No. 3, with a precursor mass ratio of 1:1 for Ni and Sn) exhibited the best catalytic performance and could be used for highly sensitive detection of formaldehyde with a detection limit as low as 2.8 nM.

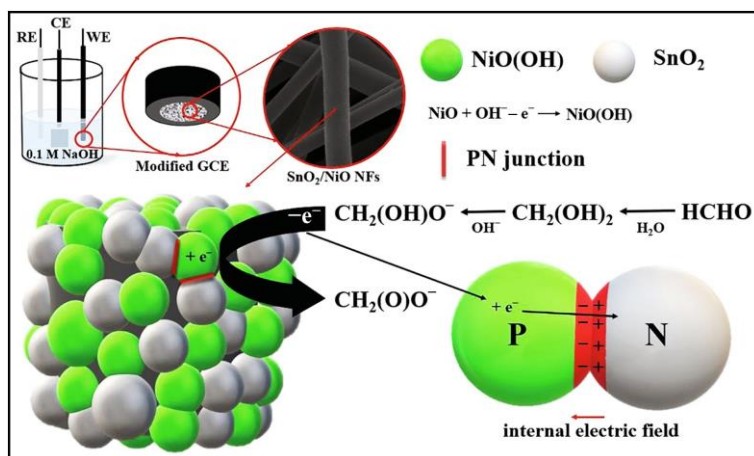

**Figure 9.** Schematic illustration of the mechanism for formaldehyde detection using a NiO/SnO$_2$ catalyst-based electrochemical sensor. Reproduced with permission [132]. Copyright 2022 Elsevier.

- Copper-based electrocatalysts

Copper exhibits various oxidation states, including Cu, Cu(I), Cu(II), and Cu(III), during the electrochemical process, which can act as effective electron-conducting intermediates for the oxidation of formaldehyde. Consequently, copper is widely employed in the construction of formaldehyde electrochemical sensors. Abnosi et al. investigated the electrocatalytic oxidation behavior of formaldehyde on a copper electrode in alkaline solutions [135]. Their cyclic voltammetry studies revealed that the presence of formaldehyde resulted in an increase in peak current associated with Cu(III) oxidation, followed by a corresponding decrease in cathodic current. Subsequently, Lin et al. prepared copper oxide (CuO) nanoparticles on a glassy carbon electrode surface using an electrodeposition method [136]. Electrochemical tests demonstrated a quasi-reversible Cu(II)/Cu(III) redox process, with an increase in anodic peak current as formaldehyde concentration increased, accompanied by the disappearance of the cathodic peak current. With increasing formaldehyde concentration, the peak potential shifted slightly in the positive direction, indicating effective electro-oxidation of formaldehyde. Momeni et al. synthesized and characterized copper oxide nanoparticles (CuO/Cu$_2$O NPs) using a green, cost-effective method in the presence of Arabic gum as a stabilizer [138]. These synthesized CuO/Cu$_2$O nanoparticles were then used to modify a carbon ionic liquid electrode (CILE), resulting in the CuO/Cu$_2$O/CILE electrode. This modified electrode exhibited effective catalytic activity towards formaldehyde oxidation. Cyclic voltammetry tests also indicated that Cu(III) generated on CuO/Cu$_2$O/CILE during the anodic scan could serve as effective intermediates for formaldehyde oxidation, resulting in a substantial increase in oxidation peak current.

Other studies have suggested that Cu(II) also plays a crucial role as a mediator in the oxidation of formaldehyde. Farhadi et al. prepared copper-porous silicon nanocomposite materials (Cu/PS) and explored their application in the detection of formaldehyde in electrochemical sensing [139]. They first deposited copper nanoparticles onto etched porous silicon (PS) surfaces using an electrodeposition method, resulting in Cu/PS nanocomposite materials. Cu NPs/PS/SPCE exhibited significant electrocatalytic activity for the

oxidation of formaldehyde at negative potentials. Compared to an unmodified electrode, the peak potential shifted approximately 0.7 V in the negative direction. Electrochemical tests suggested that the oxidation peak on Cu NPs/PS/SPCE was mainly associated with direct formaldehyde oxidation. Cu(II) ions generated during the forward scan on Cu NPs/PS/SPCE could effectively serve as intermediates for formaldehyde oxidation, resulting in a substantial increase in oxidation peak current. Tang et al. investigated the fabrication of a CuO/Cu/TiO$_2$ nanotube array (TNA)-modified electrode and its performance in formaldehyde detection [141]. The CuO/Cu/TNA-modified electrode, crafted through an electrochemical approach, showed remarkable performance in catalyzing formaldehyde's electrocatalytic oxidation (Figure 10). Initially, copper was deposited onto the TNA electrode via a pulsed current technique. This was followed by the electrode's in situ conversion to copper oxide (CuO) in an alkaline environment, a process facilitated by cyclic voltammetry. This transformation led to the CuO/Cu/TNA electrode displaying exceptional catalytic capabilities. The proposed oxidation mechanism of formaldehyde on this electrode involved the transition of Cu(II) to a more oxidized state, Cu(III). During this process, the CuO nanoparticles were inclined to further oxidize into CuOOH, a highly active species. This resulted in the creation of additional sites for catalysis, thereby boosting the electrode's overall catalytic effectiveness. The electrode's analytical attributes were impressive, demonstrating a high current density for oxidation and robust stability over multiple cycles. Its ability to detect formaldehyde spanned a broad concentration range, from 65.0 μM to 7.80 mM, with a low detection threshold of 25.0 μM. When compared to a glassy carbon electrode layered with CuO nanoparticles, the CuO/Cu/TNAs variant showed superior oxidation current density and enhanced stability during cycling, underscoring its improved efficacy in formaldehyde oxidation.

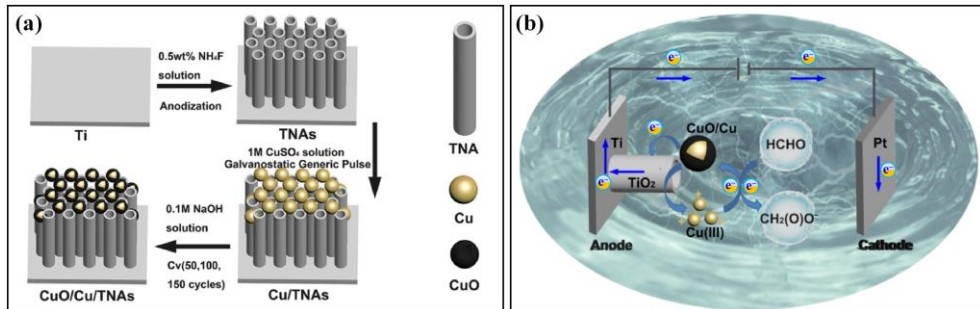

**Figure 10.** (**a**) Schematic diagram of the fabrication process of the CuO/Cu/TNA electrode; (**b**) schematic representation of the sensor for electrocatalytic oxidation of formaldehyde. Reproduced with permission [141]. Copyright 2020 Elsevier.

Other transition metal oxides or sulfides, such as TiO$_2$/RuO$_2$ [144], MnO$_2$ [145], CeO$_2$ [149], Co(OH)$_2$ [147], MoO$_x$ [146], ZnO [150,151], and Ag$_2$S [148], have also demonstrated the ability to catalyze the oxidation of formaldehyde. Bertazzoli et al. investigated the kinetics of formaldehyde oxidation in a flow electrochemical reactor with a TiO$_2$/RuO$_2$ anode [144]. The reactor employed a titanium electrode coated with (TiO$_2$)$_{0.7}$(RuO$_2$)$_{0.3}$ and monitored the electrochemical degradation of formaldehyde solutions. The oxidation of formaldehyde, as well as the removal of total organic carbon (TOC) and chemical oxygen demand (COD), was primarily controlled by mass transfer. For a solution containing 0.4 g·L$^{-1}$ of formaldehyde, the electrochemical degradation followed pseudo-first-order kinetics. The TiO$_2$/RuO$_2$ anode combination exhibited a higher rate of formaldehyde and formic acid oxidation compared to electrodes containing IrO$_2$. Nakayama et al. electrochemically deposited MnO$_2$ thin films onto glassy carbon electrodes using cathodic reduction from a KMnO$_4$ solution [145]. The MnO$_2$ was of the hollandite-type structure and demonstrated catalytic oxidation of formaldehyde under mild pH conditions (pH 6.3 or 4.0). Kim and Alba-Rubio et al. developed a sophisticated formaldehyde gas sensor using conductive carbon (Vulcan XC Max 22) treated with acid to add oxygen groups,

enabling the attachment of the molybdenum precursor, cycloheptatriene molybdenum tricarbonyl $(C_7H_8)Mo(CO)_3$ (Figure 11) [146]. This precursor was transformed into $MoO_x$ nanoparticles via surface organometallic chemistry (SOMC) and combined with carbon to create a $MoO_x$/carbon nanocomposite (Figure 11a). This composite was then applied to an SPGE electrode and covered with a Nafion layer as an ionic electrolyte. Leveraging the properties of $MoO_x$ as an n-type semiconductor with unique octahedral units, this sensor showed a high affinity for formaldehyde, distinguished by specific hydrogen bonding and nucleophilic interactions. The result was a highly sensitive and selective sensor, responding progressively to increasing formaldehyde concentrations with a low detection limit of 60 ppb and a sensitivity of 5.13 $\mu A \cdot ppm^{-1}$, effectively differentiating formaldehyde from other volatile organic compounds (VOCs).

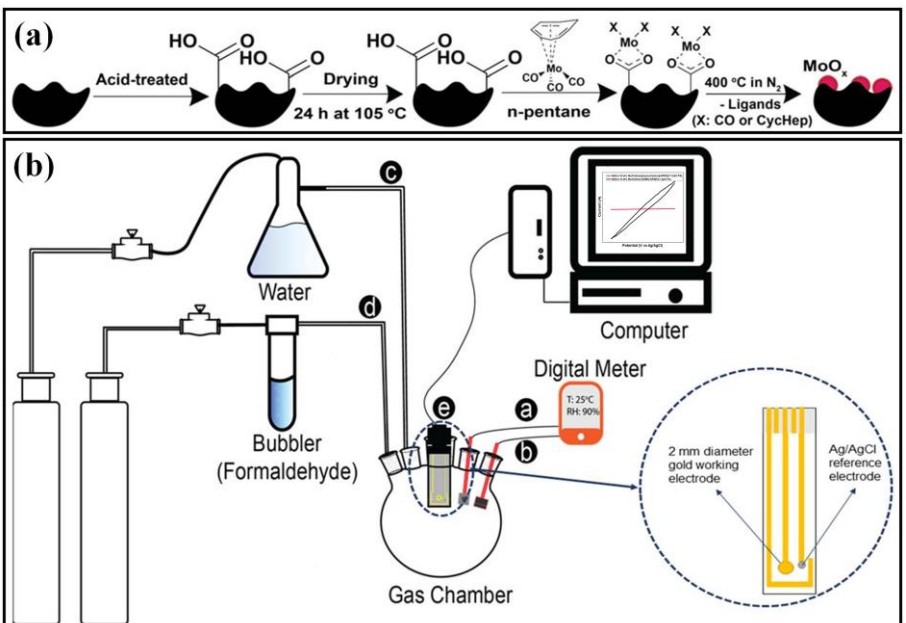

**Figure 11.** (**a**) Schematic representation of the preparation process of $MoO_x$/carbon nanocomposites. (**b**) Schematic diagram of the device for formaldehyde gas detection. Reproduced with permission [146]. Copyright 2021 The Electrochemical Society.

### 3.2.4. Organic-Polymer-Electrocatalysts-Based Formaldehyde Sensors

Organic polymer electrocatalysts, typically formed by covalent bonds in organic structures, feature large conjugated systems, providing excellent electrical conductivity and stability [176–179]. These cost-effective polymers are easily shaped and made through electrochemical polymerization or chemical redox methods. This process forms active sites on their surface or interior, enhancing their catalytic activity, particularly in oxidizing formaldehyde. Notable examples include polypyrrole [152], polyaniline [155], polydopamine [153], and polyacrylonitrile [154]. Advancements in synthesis and the exploration of new materials are expected to further improve these catalysts' efficiency and broaden their applications.

Pradhan et al. reported a molecularly imprinted polymer electrochemical sensor for the determination of formaldehyde content in food [154]. The sensor utilized polyacrylonitrile as the main component and was prepared by embedding it on a graphite electrode. The sensor fabrication process involved copolymerization of formaldehyde as the template molecule with monomers and cross-linkers, followed by template removal to leave behind the molecularly imprinted polymer. The detection principle of the sensor is based on the specific interaction and adsorption of formaldehyde molecules with the molecularly imprinted polymer on the sensor surface. When formaldehyde molecules are present, they undergo specific recognition and adsorption on the sensor's molecularly imprinted

polymer, leading to changes in the electrochemical signal. The sensor exhibited excellent electrochemical performance in NaOH solution, featuring a wide linear range from 10 μM to 1000 μM, a detection limit of 0.63 μM, and good repeatability, reproducibility, and long-term stability. The sensor was successfully applied to determine formaldehyde content in mushroom and fresh fish extracts, with results highly consistent with HPLC analysis, achieving an accuracy rate of 99%.

Varghese et al. prepared a dopamine-modified electrode (pDA/SS) using stainless steel as the substrate through dopamine electropolymerization [153]. The electropolymerization of dopamine formed a thin dopamine film on the electrode surface. This electrode was employed for formaldehyde detection, where formaldehyde undergoes oxidation in acidic aqueous solutions (Figure 12). The electrochemical method relies on monitoring the oxidation peak current of formaldehyde at different potentials to determine its concentration. The electrode exhibited high sensitivity for formaldehyde detection under optimized conditions and was capable of detecting formaldehyde in acidic aqueous solutions. The linear dynamic range of detection ranged from 0.43 μM to 1.60 μM, with a detection limit of 0.14 μM. The electrode demonstrated good reproducibility, stability, and selectivity and was successfully applied for formaldehyde detection in fish samples.

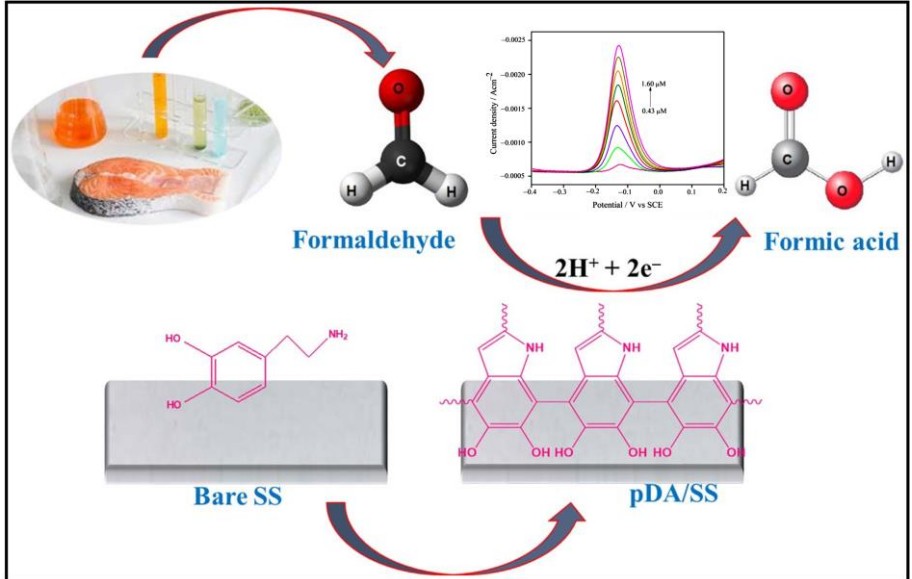

**Figure 12.** Schematic representation of the preparation and electrochemical response of the pDA/SS electrode for formaldehyde [153]. Copyright 2021 The Electrochemical Society.

### 3.3. Electrochemical Sensors Rely on Derivative Reagents

Formaldehyde, as an active organic molecule, exhibits specificity in its reactions with various organic functional groups, primarily due to its electrophilic nature and the presence of its carbonyl group. These specific reactions include the Lindlar reaction, which involves the formation of Schiff bases through the reaction of formaldehyde with amino compounds, the 2-aza-Cope reaction that leads to the rearrangement of imine structures, aldol reactions where formaldehyde adds to other carbonyl groups, and amine–formaldehyde reactions that result in the generation of compounds like urea. Leveraging these distinctive reactions, especially when combined with appropriate derivatization reagents, enables the development of highly selective formaldehyde electrochemical sensors (Table 3). Dai and Chen et al. introduced an electrochemical impedance sensor based on biomimetic electrospun nanofibers for the detection of formaldehyde [57]. They prepared polymer nanofibers (MAH/CNTs-NFs) by blending poly(methacryloyl hydrazide) (PMAH) with carbon nanotubes (CNTs) using electrospinning and used them to fabricate the working electrode (MAH/CNTs-NFs/GCE) (Figure 13). The molecular recognition sites (hydrazide) on the PMAH/CNT nanofibers can react with formaldehyde molecules to form Schiff bases,

increasing the electron transfer resistance at the electrode surface. This reaction was utilized for the detection of formaldehyde through electrochemical impedance spectroscopy (EIS). Under optimized conditions, the sensor exhibited a linear response range of 1 μM–10 mM with a detection limit of 0.8 μM. The sensor demonstrated excellent anti-interference capability and stability. Utilizing the reaction of formaldehyde with hydrazine to form hydrazones, Menart et al. introduced another electrochemical sensor for the detection of formaldehyde gas [180]. They modified screen-printed electrodes with hydrazinium polyacrylate (HPA) to create a working electrode. The modified HPA material served both as a means for the enrichment and derivatization of formaldehyde and as an electrolyte, enabling electrochemical measurements. Formaldehyde can accumulate with HPA material and form a compound called hydrazone. This hydrazone can then be reduced electrochemically to generate a current signal. The signal associated with formaldehyde occurred in a specific potential range, with optimal performance achieved in the potential scan range of –1.0 V to 1.0 V. The sensor's performance was comprehensively evaluated, and it exhibited a detection range of 4–16 ppm for formaldehyde concentration.

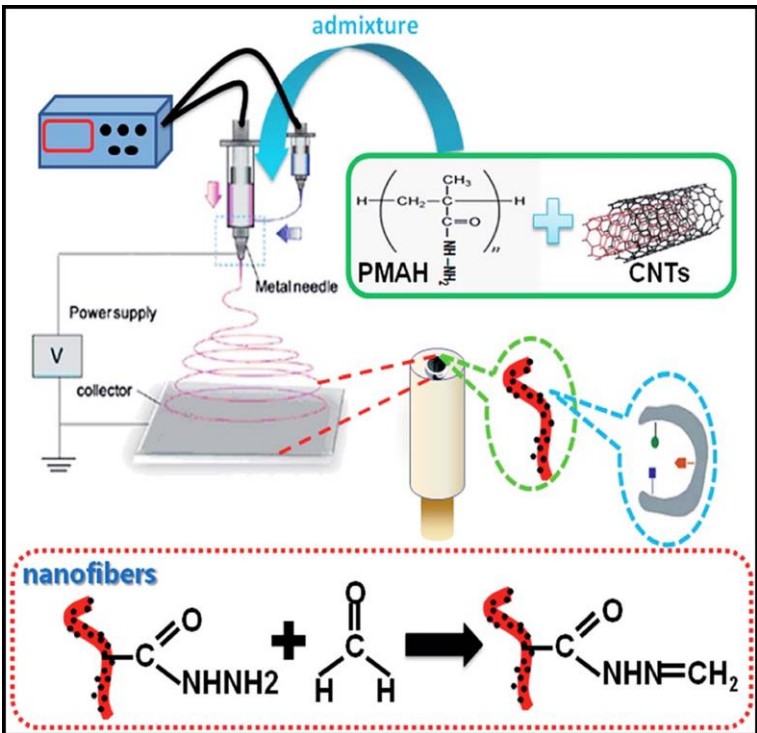

**Figure 13.** Illustration of the preparation process of PMAH/CNTs-NFs and their schematic reaction with formaldehyde. Reproduced with permission [57]. Copyright 2015 the Royal Society of Chemistry.

Li et al. reported an electrochemical impedance sensor for the detection of formaldehyde, leveraging the characteristic of formaldehyde to selectively reduce silver ions to form silver nanoparticles [181]. They prepared uniform $SiO_2$ microspheres using an improved Stöber method, followed by exchanging silver ions with trapped ammonium ions to create $Ag^+$-$SiO_2$ microspheres. These $Ag^+$-$SiO_2$ microspheres were then employed to modify a glassy carbon electrode, forming a functional electrode. Formaldehyde, acting as a reducing agent, could selectively reduce Ag+ ions to generate silver nanoparticles. As the concentration of formaldehyde increased, the proportion of silver nanoparticles increased, leading to a decrease in the interface charge transfer resistance. EIS was utilized to record changes in resistance, enabling the detection of formaldehyde concentration.

Earlier research demonstrated that formaldehyde reacts with acetylacetone and ammonia to form a yellow substance, 3,5-diacetyl-1,4-dihydromethylpyridine (DDL), a reaction historically utilized for formaldehyde detection via spectrophotometry. In a novel approach,

Silva found that DDL undergoes oxidation on unaltered glassy carbon electrodes, with an oxidation peak at 0.8 V, where formaldehyde is not electroactive (Figure 14) [58]. This discovery suggests the potential for indirectly and selectively detecting formaldehyde electrochemically. The method showed a linear detection range from 0.4 to 40.0 mg·L$^{-1}$ and a low detection limit of 0.13 mg·L$^{-1}$, proving especially useful for quickly identifying formaldehyde in diverse samples. Expanding on this, Ramos et al. observed similar electrochemical activity of DDL on standard screen-printed carbon electrodes, resulting in a characteristic oxidation peak at 0.4 V, enabling the sensitive and selective measurement of formaldehyde in wood-based products [182].

**Table 3.** Electrochemical sensors rely on derivative reagents.

| Derivative Reagents | Signal Mode | Dynamic Range | Detection Limit | Applications | Ref. |
|---|---|---|---|---|---|
|  **nanofibers** | EIS | 1 μM–10 mM | 0.8 μM | -- | [57] |
|  | Amperometry | 4–16 ppm | sub-ppm | Gas | [180] |
|  | EIS | 0.05–1 μg·mL$^{-1}$ | 41 ng·mL$^{-1}$ | Gas | [181] |
|  | Amperometry | 0.4–40.0 mg·L$^{-1}$ | 0.13 mg·L$^{-1}$ | Mushrooms | [58] |
| | Amperometry | 15–500 μM | 0.57 mg·kg$^{-1}$ | Wood | [182] |
|  | Amperometry | 0.12–1000 μM | 48.2 nM | Blood, Cell | [59] |

The Chang group proposed a novel strategy combining the 2-aza-Cope rearrangement reaction with β-elimination [183]. In this method, various hydroxy- or amino-containing molecules were integrated into FA probes using the 2-aza-Cope rearrangement combined with the β-elimination strategy [184,185]. Huang et al. designed and synthesized two electrochemical probe molecules, namely FOLP and HFOLP, using ferrocene as the electroactive moiety (Figure 15) [108]. The high electrophilic activation of the alkene group in response to formaldehyde triggers a nitrogen-heteroatom-cyclohexene rearrangement and subsequent hydrolysis and β-elimination to produce free N-alkylated aminoferrocene (AAF) as the reporter group. The N in HFOLP is connected to the electron-withdrawing carbonyl group

(acylamide), resulting in its oxidation potential at a higher value (0.22 V vs. Ag/AgCl). Upon reaction with formaldehyde, the amine is removed, leading to the appearance of the oxidation potential of the product AAF in the negative region (−0.1 V vs. Ag/AgCl). Based on this principle, high-selectivity responses to formaldehyde were achieved. The FOLP probe exhibited high sensitivity, hydrophilicity, and a low detection limit (48.2 nM). This probe demonstrated excellent selectivity, accurately detecting formaldehyde concentrations even in the presence of other interfering substances. Moreover, the FOLP probe could directly measure formaldehyde in complex physiological samples such as whole blood and cells without the need for sample pretreatment. Since formaldehyde is a byproduct of biochemical reactions for detecting creatinine, the research team successfully developed a method using the FOLP probe for the detection of creatinine, particularly in saliva. This method is characterized by its rapid, simple, non-invasive, and easy-to-operate nature, making it suitable for early diagnosis of kidney function and muscle diseases.

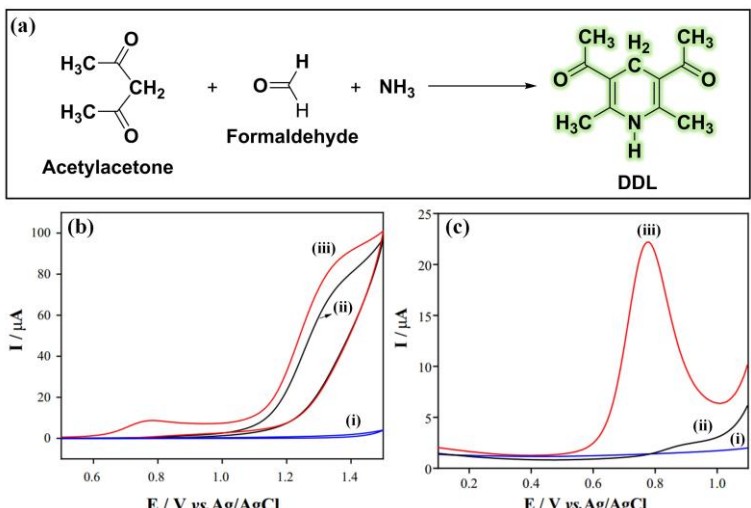

**Figure 14.** (**a**) Formation of 3,5-diacetyl-1,4-dihydropyridine (DDL) from the reaction of formaldehyde with acetylacetone and ammonia. (**b**) Cyclic voltammograms of formaldehyde (i), acetylacetone (ii), and a mixture of acetylacetone with DDL (iii). (**c**) Square-wave voltammograms of formaldehyde (i), acetylacetone (ii), and DDL (iii). Reproduced with permission [58]. Copyright 2019 Elsevier.

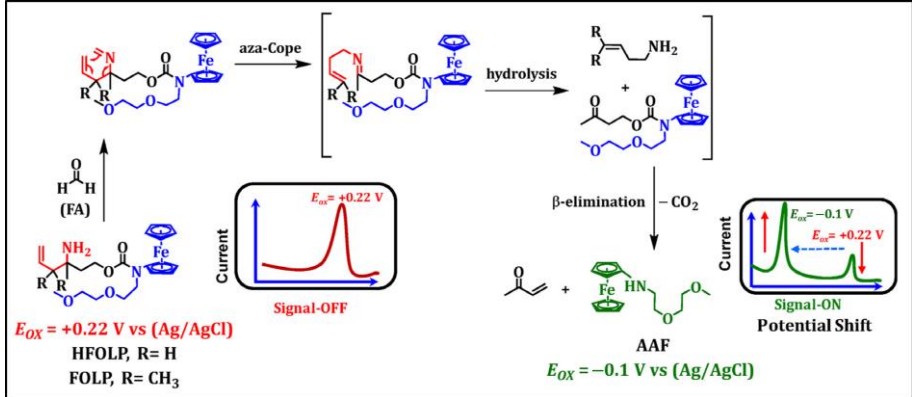

**Figure 15.** Schematic representation of the response of electrochemical probes FOLP and HFOLP to formaldehyde and the corresponding signal changes. Reproduced with permission [59]. Copyright 2021 Elsevier.

## 4. Summaries

### 4.1. Conclusions

This comprehensive review emphasizes the significant progress and diverse methodologies in the electrochemical detection of formaldehyde. Due to the need for precise,

sensitive, and rapid monitoring across various applications, the field of electrochemical detection of formaldehyde has experienced considerable growth.

(1) Diversity in Sensing Mechanisms: electrochemical sensors for formaldehyde detection are developed based on diverse principles like enzymatic reactions, usage of electrocatalysts, and specific chemical reactions, each offering unique advantages in terms of sensitivity, selectivity, and scope of application.

(2) Enzymatic Sensors: Primarily utilizing FDH, these sensors display high specificity and sensitivity towards formaldehyde. Advances in electrode material engineering and enzyme immobilization have notably enhanced their performance.

(3) Electrocatalyst-Based Sensors: Employing metals, metal oxides, and bimetallic nanocatalysts, these sensors are promising due to their high electrocatalytic activity and stability. Innovations in nanostructuring and surface modification have improved sensitivity and selectivity in formaldehyde detection.

(4) Chemical-Reaction-Based Sensors: Utilizing formaldehyde's specific chemical reactivity, these sensors offer a highly selective detection method. Success depends on the precise selection of derivatization reagents and understanding underlying chemical interactions.

### 4.2. Challenges and Future Directions

(1) Enzymatic Sensors: develop chemical alternatives to $NAD^+$ cofactors for easy electrode modification, or new electrodes that do not require cofactors, simplifying the detection system for in situ analysis of various samples.

(2) Electrocatalyst-Based Sensors: Develop finely structured electrocatalysts, like single-atom catalysts, to optimize formaldehyde oxidation efficiency and specificity. Detailed exploration of oxidation mechanisms can guide the preparation of more efficient catalysts.

(3) Chemical-Reaction-Based Probes: Expand the variety of electroactive molecules and connect diverse formaldehyde recognition groups. Drawing inspiration from well-established formaldehyde fluorescent probes could lead to a wider range of effective electrochemical probe molecules.

(4) Sensor Design: consider effective combinations of different reaction mechanisms, such as merging electrocatalyst-based sensors with specific chemical reactions for formaldehyde, to enhance selectivity and ensure rapid response.

(5) Microelectrode Technology: integration of microelectrode technology with standard electrochemical formaldehyde detection strategies could facilitate in situ real-time monitoring in biological environments.

**Author Contributions:** Conceptualization, S.C. and W.C.; methodology, Y.Y., Y.H. and L.H.; formal analysis, Y.L.; investigation, Y.Y. and Y.H.; resources, Y.H. and S.C.; writing—original draft preparation, Y.Y., Y.H., S.C. and W.C.; writing—review and editing, M.X. and W.C.; visualization, Y.Y., Y.H. and Y.L.; supervision, S.C. and W.C.; project administration, S.C. and W.C.; funding acquisition, Y.H., S.C., M.X. and W.C. All authors have read and agreed to the published version of the manuscript.

**Funding:** This work was supported by Hunan Provincial Natural Science Foundation of China (Nos. 2022JJ20052 and 2022JJ50015) and the National Natural Science Foundation of China (Project Nos. 52073087 and 22074089).

**Data Availability Statement:** Not applicable.

**Conflicts of Interest:** The authors declare no conflicts of interest.

### Abbreviations

| | |
|---|---|
| 2D | Two-dimensional |
| ALDHs | Aldehyde dehydrogenases |
| AOX | Alcohol oxidase |
| BDMA | 1,4-Benzenedimethaneamine functionalized graphene |
| BP | Buckypaper |

| | |
|---|---|
| CFP | Carbon fiber paper |
| Ch | Choline |
| CHIT | Chitosan |
| CILE | Carbon ionic liquid electrode |
| CNF | Carbon nanofiber |
| CNTs | Carbon nanotubes |
| COD | Chemical oxygen demand |
| CPE | Carbon paste electrode |
| CV | Cyclic voltammetry |
| CYPs | Cytochrome P450 |
| DDL | 3,5-Diacetyl-1,4-dihydromethylpyridine |
| DET | Direct electron transfer |
| DFT | Density functional theory |
| EDTA | Ethylenediaminetetraacetic acid |
| EIS | Impedance spectroscopy |
| EQCM | Electrochemical quartz crystal microbalance |
| ESPB | Self-powered biosensor |
| FI | Flow injection |
| FTIR | Fourier-transform infrared spectroscopy |
| GCE | Glassy carbon electrode |
| GMA-co-MTM | Glycidyl methacrylate-co-3-methylthienyl methacrylate |
| HOPG | Highly oriented pyrolytic graphite |
| HPA | Hydrazinium polyacrylate |
| IDGE | Interdigitated gold electrodes |
| IL | Ionic liquid |
| ITO | Indium tin oxide |
| LDA | Linear discriminant analysis |
| LIG | Laser-induced graphene |
| LOD | Detection limit |
| MiPAN | Molecular imprinted polymer of acrylonitrile |
| MIPs | Molecularly imprinted polymers |
| MWCNTs | Multi-walled carbon nanotubes |
| $NAD^+$ | Nicotinamide adenine dinucleotide |
| nBA-NAS | n-Butyl acrylate-*N*-acryloxysuccinimide |
| NCs | Nanoclusters |
| NF | Nickel foam |
| NFC | Near-field communication |
| NFs | Nanofibers |
| Ni/P-CPE | Nickel-doped P nanozeolite carbon paste electrode |
| NPs | Nanoparticles |
| NQS | 1,2-Naphthoquinone-4-sulfonic acid |
| NS | Nanosheets |
| NW | Nanowire |
| PAA | Polyacrylic acid |
| PAD | Amperometric detection |
| PAN | Polyaniline |
| PANI | Polyaniline |
| PB | Prussian blue |
| PBA | 1-Pyrenebutyric acid |
| pDA | Polydopamine |
| PDOS | Projected Density of States |
| PdPs-CMs | Palladium particles and carbon microspheres |
| pHEMA | Poly(2-hydroxyethyl methacrylate) |
| PMAH | Poly(methacryloyl hydrazide) |
| PMG | Poly-methylene green |
| POs-EA | Os(bpy)2-poly(vinylpyridine) |
| PPy | Polypyrrole |
| PS | Porous silicon |

| | |
|---|---|
| Pt NPs-SPUME | Pt nanoparticles-screen-printed carbon ultramicroelectrode |
| PVA | Poly(vinyl alcohol) |
| Px | p-Xylylenediamine |
| SBA-16 | Mesoporous silica Santa Barbara Amorphous No. 16 |
| SCE | Saturated calomel electrode |
| SOMC | Surface organometallic chemistry |
| SPCE | Screen-printed carbon electrode |
| SPCPtEs | Screen-printed platinized carbon electrodes |
| SPE | Screen-printed electrode |
| SSAO | Semicarbazide-sensitive amine oxidase |
| TNAs | TiO$_2$ nanotube arrays |
| TOC | Total organic carbon |
| VOCs | Volatile organic compounds |
| WGE | Paraffin-impregnated graphite electrode |

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
