# Peer review of "Recent Advances in Electrochemical Sensors for Formaldehyde"

_molecules, doi:10.3390/molecules29020327_

Round 1

Reviewer 1 Report

Comments and Suggestions for Authors

 The present manuscript review the sensing strategies for electrochemical detection of formaldehyde. Considering the importance of formaldehyde detection in various samples using sensitive and selective platforms, the topic is expected to be of interest for the readers. There are only some slight modifications needed to be addressed as follows:

- Please provide necessary and relevant references wherever needed such as in the case of "There have been reports of electrochemical sensors based on formaldehyde derivatization reactions."

- Please make sure that the abbreviations are mentioned via their first indication in the text. There are many repetitions in abbreviations and some of them are not addressed appropriately in the text.

- Table 2, raw 8 should be corrected.

- Figures 4 and 9: The captions need correction.

- Figures 8 and 11 are not addressed in the text. 

Author Response

Reviewer 1

The present manuscript review the sensing strategies for electrochemical detection of formaldehyde. Considering the importance of formaldehyde detection in various samples using sensitive and selective platforms, the topic is expected to be of interest for the readers. There are only some slight modifications needed to be addressed as follows:

Comment 1: Please provide necessary and relevant references wherever needed such as in the case of "There have been reports of electrochemical sensors based on formaldehyde derivatization reactions.

Response: We sincerely appreciate the valuable feedback from the reviewer. We have now added relevant references after the statement "There have been reports of electrochemical sensors based on formaldehyde derivatization reactions [57-59].".

Comment 2: Please make sure that the abbreviations are mentioned via their first indication in the text. There are many repetitions in abbreviations and some of them are not addressed appropriately in the text.

Response: We have re-examined the abbreviations in the manuscript and added the corresponding full forms upon their initial appearances in the text. Examples include Alcohol oxidase (AOX, EC 1.1.3.13), MiPAN@GP (molecular imprinted polymer of acrylonitrile@graphite paste), pDA (polydopamine), SPCE (screen-printed carbon electrode), HOPG (highly oriented pyrolytic graphite), and EDTA (ethylenediaminetetraacetic acid). Due to space constraints, a list of abbreviations used in tables has been included at the end of the main text.

Comment 3: Table 2, raw 8 should be corrected

Response:  We have corrected the entry for "Au electrode" in that specific position.

Comment 4: Figures 4 and 9: The captions need correction.

Response: We have revised the captions for Figures 4 and 9.

Comment 5: Figures 8 and 11 are not addressed in the text.

Response: We have appropriately referenced Figures 8 and 11 in the revised manuscript.

Reviewer 2 Report

Comments and Suggestions for Authors

The proposed paper presents an excellent review of formaldehyde determination methods using electrochemical detection. The authors gave a detailed review of the literature that besets this problem and the ways that researchers have proposed to achieve the best possible analytical parameters of their methods. In order for the work to be accepted, a minor revision is required.

1. it is necessary to give a graphic representation of the trend in the number of papers in the last ten years on this topic, where SCOPUS or google scholar would serve as a source.

2. 13 out of almost 150 references from 2023 raise the question of whether this is a recent trend. It is necessary to refresh the references as young as possible

Author Response

Reviewer 2

The proposed paper presents an excellent review of formaldehyde determination methods using electrochemical detection. The authors gave a detailed review of the literature that besets this problem and the ways that researchers have proposed to achieve the best possible analytical parameters of their methods. In order for the work to be accepted, a minor revision is required.

Comment 1: it is necessary to give a graphic representation of the trend in the number of papers in the last ten years on this topic, where SCOPUS or google scholar would serve as a source.

Response: We sincerely appreciate the reviewer's evaluation and suggestion. We have compiled a graphical representation illustrating the recent trend in the number of publications on electrochemical methods for formaldehyde detection in the past ten years. The data were sourced from Web of Science, and the graphic representation is provided in Scheme 1.

Scheme 1. Statistical analysis of the number of papers published on formaldehyde electrochemical sensors in the past 20 years. Data were retrieved from Web of Science using the search terms "Electrochemical Sensors" and "Formaldehyde" as search topics. Search date: 1/5/2024.

Comment 2: 13 out of almost 150 references from 2023 raise the question of whether this is a recent trend. It is necessary to refresh the references as young as possible.

Response: We have conducted a thorough review of the literature and made efforts to incorporate the latest publications into our manuscript. As of now, the number of references from 2023 and beyond has increased to 20, with the majority of references dating from 2020 onwards. We believe that the current version adequately reflects the most recent advancements and dynamics in the field. We appreciate your valuable suggestion.
